# A Pattern Classification Distribution Method for Geostatistical Modeling Evaluation and Uncertainty Quantification

**Chen Zuo** [1], **Zhuo Li** [1], **Zhe Dai** [1,*], **Xuan Wang** [2] and **Yue Wang** [3]

1. Department of Big Data Management and Applications, Chang'an University, Xi'an 710064, China; chenzuo@chd.edu.cn (C.Z.); zhuoli803@chd.edu.cn (Z.L.)
2. School of Computer and Control Engineering, Yantai University, Yantai 264005, China; xuanwang91@ytu.edu.cn
3. Xi'an Key Laboratory of Digital Construction and Management for Transportation Infrastructure, Xi'an 710064, China; ywang@chd.edu.cn
* Correspondence: zhedai@chd.edu.cn

**Abstract:** Geological models are essential components in various applications. To generate reliable realizations, the geostatistical method focuses on reproducing spatial structures from training images (TIs). Moreover, uncertainty plays an important role in Earth systems. It is beneficial for creating an ensemble of stochastic realizations with high diversity. In this work, we applied a pattern classification distribution (PCD) method to quantitatively evaluate geostatistical modeling. First, we proposed a correlation-driven template method to capture geological patterns. According to the spatial dependency of the TI, region growing and elbow-point detection were launched to create an adaptive template. Second, a combination of clustering and classification was suggested to characterize geological realizations. Aiming at simplifying parameter specification, the program employed hierarchical clustering and decision tree to categorize geological structures. Third, we designed a stacking framework to develop the multi-grid analysis. The contribution of each grid was calculated based on the morphological characteristics of TI. Our program was extensively examined by a channel model, a 2D nonstationary flume system, 2D subglacial bed topographic models in Antarctica, and 3D sandstone models. We activated various geostatistical programs to produce realizations. The experimental results indicated that PCD is capable of addressing multiple geological categories, continuous variables, and high-dimensional structures.

**Keywords:** geostatistical modeling; multiple-point statistics; uncertainty quantification; subglacial topographic model; hydrological model

## 1. Introduction

Geological models play an important role in a wide range of real-world applications. Recent developments in the Earth surface dynamics have highlighted the importance of high-quality hydrological models [1]. A set of stochastic realizations comprises fundamental materials to express the spatiotemporal evolution of delta and flume systems [2,3]. Moreover, there has been increasing interest in the high-resolution subglacial topography models [4,5]. The roughness of bedrock has a substantial influence on subglacial flow behaviors in Arctic and Antarctica [6,7]. With the development of computing platforms, multiple-point statistics (MPS) has gained considerable attention [8,9]. With the aim of creating realistic models, MPS concentrates on the relationship between one target point and neighboring points. Viewing the training image (TI) as a prior material, the spatial patterns are constantly extracted and reproduced in the simulation grid (SG). With the objective of improving the simulation quality as well as computational efficiency, a range of image-processing and machine-learning techniques have been introduced into the MPS framework [10–12]. For example, spatial correlation is used to create an adaptive template and conserve patterns [13,14]. In order to save running time, clustering is a feasible way to

organize patterns in TI and find representatives [7,15–17]. The medoid of each group has a high rank in the downstream procedure. In addition, multi-grid analysis is employed to capture spatial structures across different resolutions [18,19]. During the MPS simulation, the long-range connectivity is regenerated before the fine-grain characteristics. In addition, the development of the generative adversarial network (GAN) technique has received considerable attention in the geostatistics community [20,21]. Based on a large amount of TIs, two neural networks are simultaneously trained through an adversarial competition. A generator network attempts to produce an image associated with similar characteristics to TIs. By contrast, the discriminator is responsible for distinguishing real and simulated models. The expanding applications of GAN include geological facies [22,23], probability inversion [24], and porous media [25].

One major challenge for MPS, GAN, and other TI-based modeling programs is to quantitatively evaluate simulation quality. Therefore, numerous descriptors have been presented. As a classical two-point statistics metric, variogram focuses on calculating the expected squared difference between two points divided by a certain distance [26]. By contrast, the two-point correlation function and the lineal-path function are broadly used to characterize microstructures [27]. The former concentrates on the probability that two randomly chosen points have the same material phase. The latter approach is defined as the possibility that a straight line is entirely in a certain facies. Moreover, the connectivity functions are devised to compute the probability that two points in SG belong to the same connected component [28]. While two-point approaches are used in a variety of applications, a shortcoming is that geometrically and morphologically complicated structures cannot be finely represented with these methods.

The limitations of two-point statistics motivate researchers to develop high-order methods. From the geostatistical point of view, there are two variabilities within simulated realizations: pattern reproduction and spatial uncertainty [29]. On one hand, the core task of the geostatistical simulation method is to reproduce spatial patterns in SG. This is favorable for exhibiting consistencies between TI and the generated models. On the other hand, spatial uncertainty plays an essential role in understanding Earth systems. The use of a group of stochastic realizations is helpful to represent uncertainty and randomness. Therefore, competitive methods not only create similar realizations to TI but also enrich the diversity within generated models. Furthermore, the observation variable is important prior knowledge in conditional simulations. For example, the borehole interpretation is directly sampled from the subsurface system [17]. Produced by gound penetrating radar, the geophysical data describe the trend of the geological structure under investigation [30]. It is necessary to respect conditioning data during geological modeling. These conflicting objectives create a challenge for simulation programs.

Based on the variabilities mentioned above, multiple-point histogram (MPH) is reported to assess the quality of the unconditional simulation [31]. First, the program extracts spatial patterns from a geological model. Second, a probability distribution is created according to the frequency of each pattern. Third, MPH views the difference between two pattern distributions as a measure of the distance between two geological models. In particular, Jensen–Shannon (JS) divergence is applied to distinguish two distributions. The pattern reproduction is expressed by the average distance between the TI and the simulated realizations. By contrast, the mean distance between the generated models implies the spatial uncertainty. However, one primary drawback of MPH is the ability to describe complicated structures. Within the MPH framework, the template is the key to capturing geological patterns. Since MPH records every possible pattern configuration, the dimension of the pattern distribution grows rapidly. There is a tradeoff between the running speed and the evaluation accuracy. On one hand, an extending template is useful for identifying complex structures. On the other hand, high-dimensional distributions have a negative effect on the calculational efficiency. It is time-consuming to apply large templates in MPH.

With the purpose of improving evaluation performance, the analysis of distance (ANODI) was designed by Tan et al. [32]. The technical developments were as follows:

(1) Patterns in the TI are organized by a clustering method. The medoid of each group becomes the representative instance. (2) The program classifies patterns in the SG on the basis of their distances with representative patterns. A cluster-based pattern histogram is created according to the number of members in each group. (3) Similar to MPH, JS divergence is employed to quantify the difference between two histograms. The program performs multi-dimensional scaling (MDS) to visualize the affinity between geological models. (4) A multi-grid strategy is utilized to analyze geological structures across difference scales. The program individually captures long-range structures as well as fine-grain patterns. A weighted aggregation is conducted to combine JS divergences from several resolutions.

In recent years, ANODI has been used to examine the simulation quality in various applications. However, two primary concerns are the parameter specification and the evaluation accuracy. There are three noticeable technical limitations. First, the size of the template and the number of clusters are two user-defined parameters. Inappropriate configurations bring uncorrelated and redundant knowledge into the pattern analysis. Second, the program organizes patterns in the SG by calculating their distances with the medoid patterns in the TI. Given a TI with complicated structures, a large number of prototypes are found during the clustering step. It is time-demanding to compare the patterns and every representative. The time consumption constrains the dimension of the template and pattern groups. Third, the weight of each resolution is fixed and constant in the multi-grid analysis. The intrinsic characteristics of the geological structure are not taken into account.

In this paper, we provide a valuable alternative to quantitatively evaluate geostatistical modeling and quantify uncertainty. With the objective of improving the evaluation accuracy and simplifying the parameter specification, a pattern classification distribution (PCD) program is proposed to compare geostatistical realizations. First, our program applies an irregular template of adaptive size to extract geological patterns. According to the spatial correlation in the TI, the template points are sequentially gathered by a region-growing program. The computer controls the number of conditioning points based on the elbow point of the entropy function. Second, a clustering-and-classification program is designed to characterize the geological models. Aiming to customize the parameter setting, we apply hierarchical clustering to group the training patterns. Our program applies a decision tree to classify geological patterns and creates a pattern classification distribution from a geological realization. The similarity between two geological models is defined by the JS divergence between two distributions. Third, we devise a stacking framework to develop the multi-grid analysis. To improve the aggregation accuracy, the importance of each grid is calculated according to the intrinsic characteristics of the TI. A large weight is assigned to the coarse grid when there is an intensive long-range dependency in the TI.

We conducted four practical applications with the intention of comprehensively examining the proposed method. In the first test, a benchmark channel model was utilized. We ran a range of MPS programs to generate hydrological models. MPH, ANODI, and our PCD are applied to rank the realization sets. Compared with the existing methods, the key advantage of our PCD is the automatic parameter specification according to the simulation scenario. The geological models are reasonably distinguished by the proposed method. Further applications include non-stationary flume realizations, subglacial digital elevation models in Antarctica, and three-dimensional sandstone models. PCD exhibits a versatile ability to solve multiple geological categories, continuous variables, morphologically complex structures, and high-dimensional structures.

The rest of this paper is organized as follows. Section 2 establishes the context of the geostatistical evaluation methods and provides detailed procedures within MPH and ANODI. Our proposed PCD is explained in Section 3. Section 4 presents four real-world applications. The experimental results and findings are discussed in Section 5. Finally, conclusions are drawn in Section 6.

## 2. Background of the Geostatistical Evaluation Methods

### 2.1. Multiple-Point Histogram

Prior to explaining the proposed program, we provide a brief overview of the MPH and ANODI methods. There are three basic steps within MPH [31]. (1) Based on a geological model, spatial patterns are extracted by a predefined template. (2) The program records the frequency of each pattern. The pattern histogram becomes a tool to describe TI and geological realizations. (3) Jensen–Shannon divergence is utilized to measure the similarity between two distributions.

The MPH program is illustrated in Figure 1. To simplify the explanation, a template with five points is applied. As Figure 1a shows, the program visits point $a$ and creates a pattern $p(a) = (Z(a + u_1), Z(a + u_2), Z(a + u_3), Z(a + u_4), Z(a + u_5)) = (1, 0, 1, 0, 1)$. Here, $Z(a)$ denotes the geological state of the point $a$. The program continuously visits every available point in TI. In this case, 36 patterns were found.

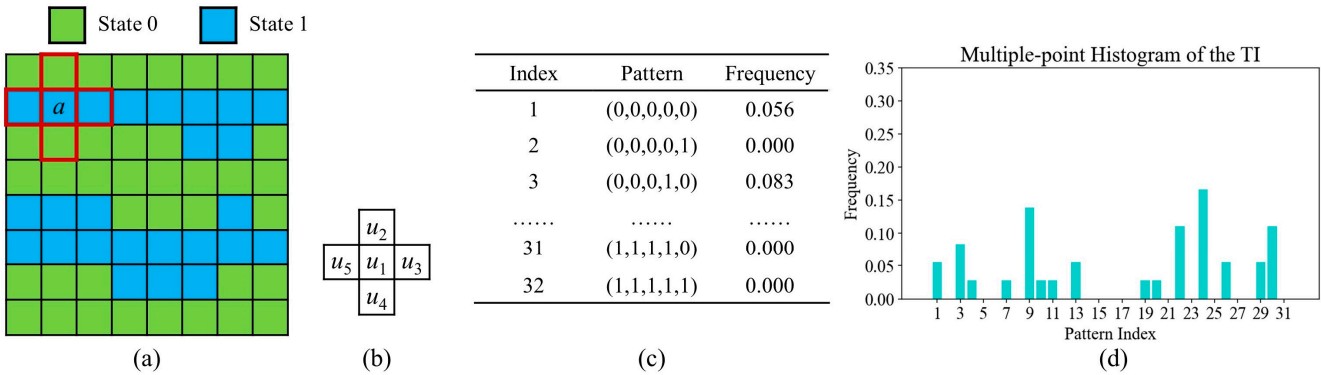

**Figure 1.** Multiple-point histogram based on a conceptual image. (**a**) A training image with size of $8 \times 8$. There are two geological states within the TI. The first pattern centered on the point $a$ is highlighted in red; (**b**) a template with five conditioning points; (**c**) pattern frequency table; (**d**) the resulting multiple-point histogram.

Next, MPH focuses on analyzing patterns and creating a descriptor. Let $f_1$ denote the frequency of pattern $p_1$. As shown in Figure 1c,d, a pattern frequency table and a histogram are produced by recording the occurrence of each pattern. The number of possible patterns is $2^5 = 32$ becuase there are five conditioning points in the template and two geological categories in TI. Accordingly, the dimension of the pattern histogram is specified as 32.

After counting the frequency of each pattern, MPH regards the distance between two distributions as a measure of the dissimilarity between two geological models. In particular, JS divergence is an appropriate way to compare two histograms [33]. Suppose that there are two distributions $P_1 = \left\{ f_1^{(1)}, f_2^{(1)}, \ldots, f_{32}^{(1)} \right\}$ and $P_2 = \left\{ f_1^{(2)}, f_2^{(2)}, \ldots, f_{32}^{(2)} \right\}$. The JS divergence is defined as follows:

$$dis_{JS}(P_1, P_2) = \frac{1}{2} \sum_{i=1}^{32} f_i^{(1)} \log \left( \frac{f_i^{(1)}}{f_i^{(2)}} \right) + \frac{1}{2} \sum_{j=1}^{32} f_i^{(2)} \log \left( \frac{f_j^{(2)}}{f_i^{(1)}} \right) \tag{1}$$

As mentioned above, there are two variabilities within the geological models: pattern reproduction and spatial uncertainty. On one hand, the simulation program repeatedly extracts structures from the TI and reproduces proper patterns in the SG. The consistency between the TI and the generated realizations plays a key role in the assessment of geological modeling quality. On the other hand, uncertainty is a fundamental factor in exploring the surface and subsurface system. In general, the stochastic simulation method creates an ensemble of realizations. The distance between two geological realizations should be sufficiently large to represent uncertainty and randomness. Within the MPH framework, two variabilities are individually measured by the mean distance. Suppose that there is

one training image *TI* and several geological realizations *RE*. The pattern reproduction ability of the geostatistical modeling method is quantified as follows:

$$dis_{RE}^{within} = \frac{1}{L} \sum_{l=1}^{L} dis_{JS}\left(P_{TI}, P_{RE}^{(l)}\right) \tag{2}$$

where $L$ is the number of geological models and $P_{RE}^{(l)}$ is the pattern histogram computed from the $l$-th realization.

By contrast, the average distance between two geological models becomes a measure of uncertainty. The computation detail is shown below:

$$dis_{RE}^{between} = \frac{1}{L(L-1)} \sum_{l=1}^{L} \sum_{l'=1}^{L} dis_{JS}\left(P_{RE}^{(l)}, P_{RE}^{(l')}\right) \tag{3}$$

Next, MPH applies the preceding distances to compare two modeling methods. Suppose that there are two realization sets, *A* and *B*. The output ratios are defined as follows:

$$r_{A,B}^{between} = \frac{dis_A^{between}}{dis_B^{between}} \tag{4}$$

$$r_{A,B}^{within} = \frac{dis_A^{within}}{dis_B^{within}} \tag{5}$$

$$r_{A,B}^{overall} = \frac{r_{A,B}^{between}}{r_{A,B}^{with}} \tag{6}$$

The first ratio $r_{A,B}^{between}$ focuses on the extent of uncertainty. A high value of $r_{A,B}^{between}$ indicates that set *A* has greater uncertainty and diversity than set *B*. In comparison, the pattern reproduction ability is quantified by $r_{A,B}^{within}$. A small value of $r_{A,B}^{within}$ reveals that set *A* has a close affinity with the TI. The last ratio $r_{A,B}^{overall}$ summarizes previous two aspects. The $r_{A,B}^{overall} > 1.0$ implies that set *A* has better quality than set *B*.

One key limitation of MPH is the tradeoff between the evaluation accuracy and the running speed. It is worth noting that MPH identifies the relationship between multiple points. For a geometrically complex structure, an extending template has a substantial effect on the characterization quality. However, a large template dramatically increases the number of possible patterns. For example, supposing that a template of $5 \times 5$ is used to analyze a TI with two geological categories, the dimension of multiple-point histogram becomes $2^{25} = 33,554,432$. It is time-consuming and memory-intensive to handle high-dimensional histograms. Furthermore, the pattern histogram becomes a sparse vector when a large template is applied. Numerous zero values not only affect the effectiveness of JS divergence but also lead to considerable computation costs. Therefore, MPH generally applies small templates. Morphologically complicated structures cannot be represented effectively.

### 2.2. Analysis of Distance

Aiming to overcome the limitations of MPH, Tan et al. proposed the analysis of distance [32]. Their core contribution was the application of k-means clustering to group training patterns. Figure 2 provides a conceptual example. Similar to MPH, the first step in ANODI is to extract spatial patterns from the TI. Using a template with five points, 36 patterns are captured. Next, the program performs k-means clustering to inspect the underlying proximity within these patterns. The instances with strong similarity are allocated into one group. Since there is a categorical variable in the TI, Hamming distance is applied to distinguish between patterns. Moreover, the computer performs multidimensional scaling (MDS) to facilitate the visualization. In MDS feature space, one node represents a pattern. Two similar patterns have a small distance in the feature

space. As shown in Figure 2e, four clusters are detected by the k-means program. Next, ANODI concentrates on finding the medoid of each group. The medoid is defined as the pattern closest to the geometrical center of a cluster. In addition, the program records the size of each cluster. As shown in Figure 2g, a cluster-based histogram of patterns (CHP) is created. The key benefit of ANODI is the dimension of the pattern distribution. It is convenient to control the dimension of the pattern histogram by specifying the number of pattern clusters. The high-dimension issue in MPH is therefore significantly alleviated.

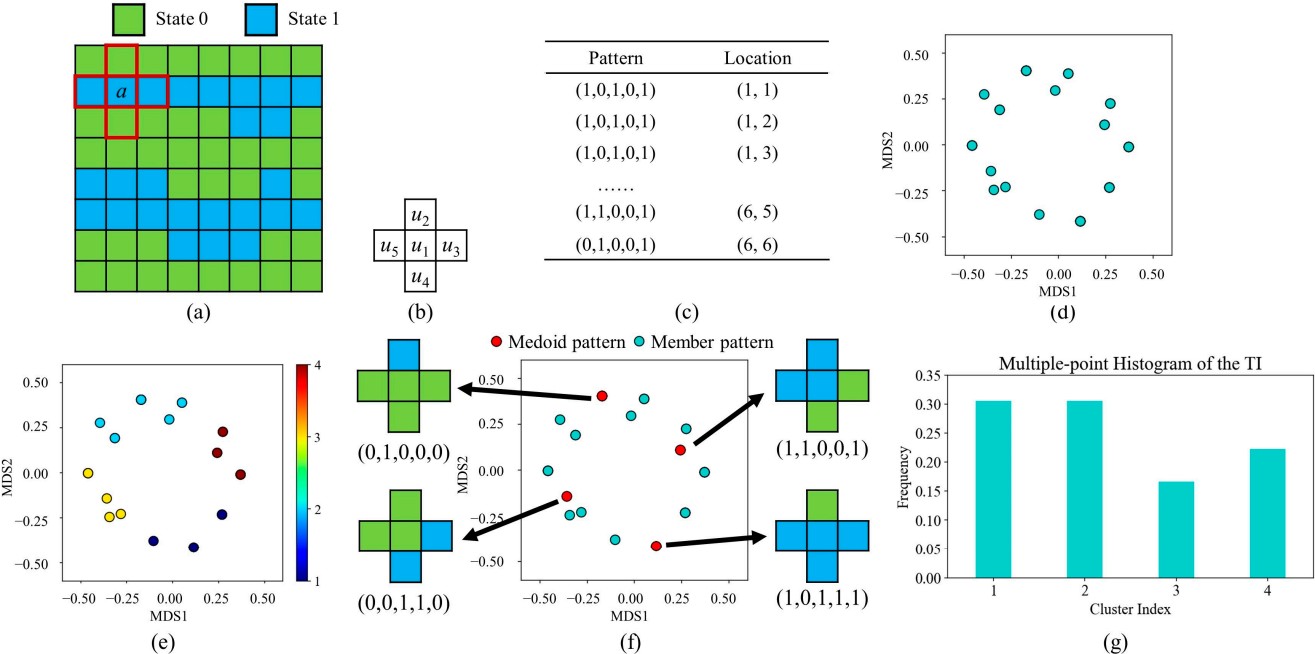

**Figure 2.** Analysis of distance based on a conceptual image. (**a**) Training image; (**b**) template; (**c**) pattern dataset. (**d**) patterns in the MDS feature space. (**e**) 4 pattern groups created by k-means clustering; (**f**) representative patterns; (**g**) cluster-based histogram created by TI.

Based on the medoid patterns, ANODI attempts to extract the morphological characteristics and rank the geological models. The characterization procedure is composed of three steps. (1) The patterns are extracted from the geological realizations. (2) The program classifies the pattern examples according to their distances from the representatives. (3) The number of member instances in each pattern category is output as a pattern histogram. Next, JS divergence is carried out to compare two distributions. Within the ANODI framework, a close similarity between two distributions suggests that there is a strong agreement between two geological realizations. Moreover, MDS is introduced to visualize the relationship between realizations. Based on a distance matrix, MDS projects data points into the low-dimensional feature space. The geological models which show matching structures are close in the MDS space. The technical details of MDS are elaborated in [1].

In addition, long-range correlations and connectivity are common in geostatistical simulations. It is difficult to extract long-scale structures in accordance with small templates. Therefore, the multi-grid strategy is incorporated into the ANODI framework. As shown in Figure 3, the program creates a pyramid of multi-resolution views. Inspired by MPS, the coarse grid is recursively generated by subsampling the fine grid. In this conceptual case, we implement a down-sampling procedure of stride 2. Starting from the bottom-left corner, the pixels in the fine grid are sequentially checked. The computer removes every even-numbered row and column to produce a small grid. For complex scenarios, a Gaussian pyramid and facies-frequency-based methods are favorable to preserve important geological structures. Let $G$ denote the number of grids. Therefore, one training image $TI$ can be expanded into a set of multi-resolution grids $\{TI_1, TI_2, \ldots, TI_G\}$. In the similar manner, a geological realization $RE$ is extended into $\{RE_1, RE_2, \ldots, RE_G\}$. Based on a

specified grid $g$, the computer performs pattern extraction, k-means clustering, and pattern classification to create a cluster-based histogram $P_{RE,g}^{(l)}$ from the realization $RE_g$. In a multi-grid context, the average distances in Equations (2) and (3) are modified as follows:

$$dis_{RE,G}^{within} = \frac{1}{L} \sum_{g=1}^{G} w_g \sum_{l=1}^{L} dis_{JS}\left( P_{TI,g}, P_{RE,g}^{(l)} \right) \tag{7}$$

$$dis_{RE,G}^{between} = \frac{1}{L(L-1)} \sum_{g=1}^{G} w_g \sum_{l=1}^{L} \sum_{l'=1}^{L} dis_{JS}\left( P_{RE}^{(l)}, P_{RE}^{(l')} \right) \tag{8}$$

where $w_g$ is the weight of the $g$-th grid. In ANODI, a fixed weight $w_g = 1/2^g$ is applied. The high-resolution images and the fine grid are assigned high contributions. There are two assumptions behind this design. First, that there is less information and variability in low-resolution grids. Second, that short-scale patterns are more important than large-scale structures.

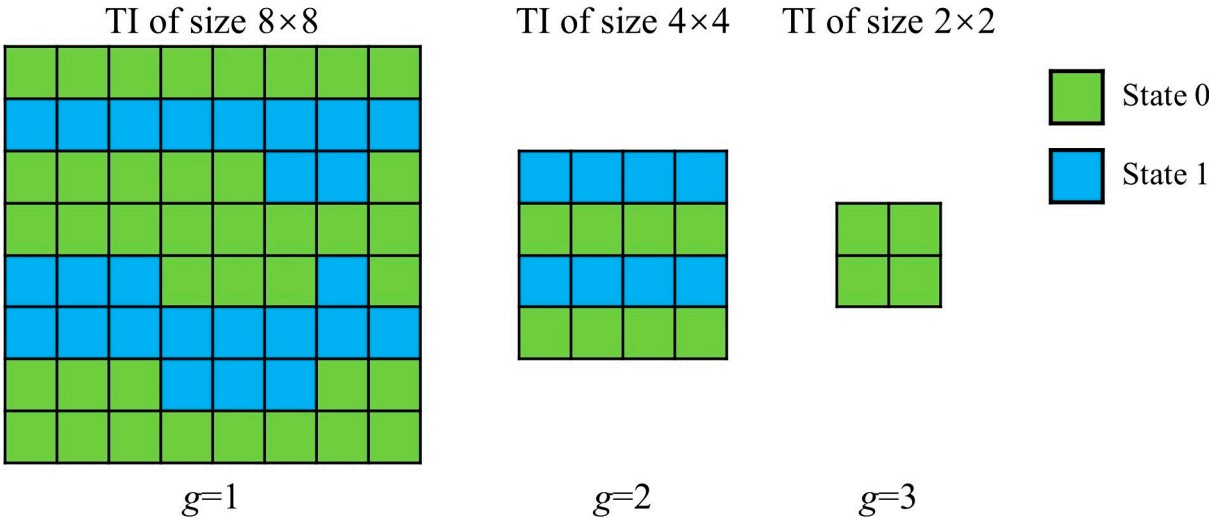

**Figure 3.** Multi-resolution TIs in the multi-grid analysis.

Although a range of practical applications are performed, there are three key technical limitations in ANODI. (1) The parameter specification is a key step to ensure the evaluation accuracy. Prior to the pattern-extraction step, the user has to specify the shape and size of template. Moreover, it is necessary to set the number of groups in the k-means clustering. An unsuitable setting has a negative influence on the computation quality. (2) It is time-consuming to generate the cluster-based histogram from the geological realization. In order to classify patterns, ANODI has to calculate the distance with every representative pattern in the TI. (3) The multi-grid analysis suffers from the fixed weight. The long-range structure and connectivity do not receive sufficient attentions.

## 3. The Key Principles of Pattern-Classification Distribution

### 3.1. The Correlation-Driven Template-Design Program

The first step in our method is to design a reasonable template to extract geological patterns. Compared with the two-point statistics method, one key benefit of MPH and ANODI is the application of a template to explore the relationship between multiple points. However, these two methods apply a fixed and user-defined template. In order to improve the modeling quality, there are many adaptive template-design methods in the MPS community. For example, Honarkhah and Caers (2010) presented a template-selection method in their distance of pattern (DISPAT) [16]. The entropy is a measure of the information needed to encode a pattern. The program finds the optimal template size using elbow-point detection. The key drawback is that the DISPAT template is always

square. The program cannot change the shape of the template according to the TI of interest. It is difficult to address anisotropic structures. By comparison, correlation-driven direct sampling (CDS) is an applicable way to quantify the contribution of each template point [14]. At first, the spatial dependency of the TI is analyzed by the correlation coefficient. With the purpose of removing the effects of noise and geologic cyclicity, CDS employs a Gaussian function to approximate the correlogram. Next, the weight of each template point is calculated by the Gaussian function. Nevertheless, CDS cannot automatically control the number of conditioning points. The template size is a user-defined parameter.

In this work, we combine the strengths of DISPAT and CDS together. An irregular template of adaptive size is devised to capture patterns. Based on the inherent characteristics of the TI, the computer automatically determines not only the shape but also the size of the template. Figure 4 provides an example to discuss our correlation-driven template design program. There are four basic steps. (1) Motivated by CDS, we compute the correlation coefficient between the template center and each neighboring point. In this conceptual case, a template with a size of $3 \times 3$ is employed to extract patterns. In Figure 4b, the deep purple indicates a strong dependency. Since there are two channels in the TI, an intensive correlation is presented in the horizontal direction. (2) A region-growing program is activated to sequentially collect template points. Viewing the template center as the seed point, we iteratively incorporate the neighboring point with the maximum correlation into the template. The template points that exhibit strong relationships with the center are given the highest priority. As shown in Figure 4c, the program creates several candidate templates with irregular shapes. (3) Our program assesses the template by means of the entropy. Based on a specified template, a group of patterns are extracted from the TI. Inspired by DISPAT, the entropy of the pattern distribution becomes a measure of the information captured by the template. We repeatedly perform steps 2 and 3 until every template configuration is examined. In this case, an entropy function of the pattern histogram is displayed in Figure 4d. (4) An elbow-point-detection technique is utilized to find the optimal parameter. In this work, we apply the profile log-likelihood approach [34]. Let $E = \{e_1, e_2, \ldots, e_N\}$ denote the entropy set. $N$ is the number of template points. For every instance of entropy, we define two groups: $\{e_1, e_2, \ldots, e_n\}$ and $\{e_{n+1}, e_{n+2}, \ldots, e_N\}$. Next, the profile log-likelihood function $l(n)$ is defined as:

$$l(n) = -n \log\left(\frac{1}{\sqrt{2\pi\sigma^2}}\right) \sum_{i=1}^{n} \frac{(e_i - \mu_1)^2}{2\sigma^2} + (n - N) \log\left(\frac{1}{\sqrt{2\pi\sigma^2}}\right) \sum_{i=n+1}^{N} \frac{(e_i - \mu_2)^2}{2\sigma^2} \quad (9)$$

$$\sigma^2 = \frac{(n-1)\sigma_1^2 + (N - n - 1)\sigma_2^2}{N - 2} \quad (10)$$

where $\mu_1$ and $\mu_2$ are the means of the two groups, respectively. In contrast, $\sigma_1$ and $\sigma_2$ are sample variances. The common scale variance is denoted by $\sigma$. Consequently, the position with the maximum values of $l(n)$ is the optimal choice. In this case, the program specifies the template size as 4.

### 3.2. Geological Model Characterization Using Hierarchical Clustering and Decision Tree

Based on the template described above, we extracted a range of patterns from the TI. A key technical problem is the organization of these training instances. In this work, our program applies the agglomerative hierarchical clustering method [35]. The advantages of this method include: (1) There is no strong assumption on the distribution of the clusters. As a bottom-up approach, the program constantly merges similar groups. The local connectivity plays an essential role in the clustering step. Thus, the use of hierarchical clustering makes it possible to tackle data groups with varying densities and affinities. (2) It is easy to address categorical variables as well as continuous variables. Our program employs Hamming distance to deal with categorical data. By contrast, continuous variables are handled by the normalized Euclidean distance. (3) Rather than the number of groups, the distance threshold becomes the user-defined parameter in the hierarchical clustering. The program recursively performs the merging step until there is no group whose distances

from others are shorter than the predefined tolerance. For a TI with diverse structures, the hierarchical clustering method automatically adopts numerous groups to organize pattern instances. In comparison, few clusters are produced when there are repetitive and redundant structures in the TI.

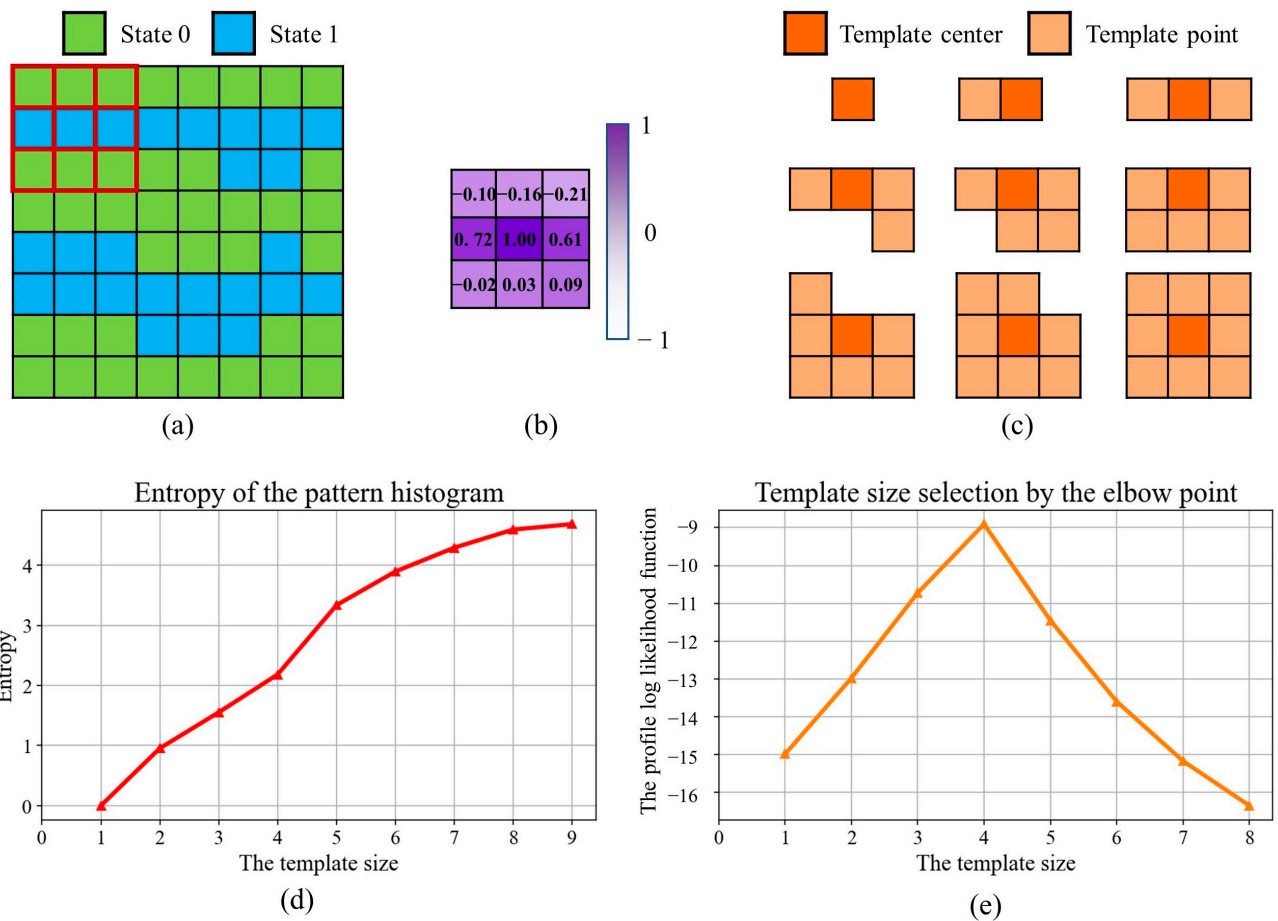

**Figure 4.** An irregular template of adaptive size. (**a**) Training image. The first pattern captured by a template of 3 × 3 is shown in red; (**b**) the correlation coefficient between the template center and each neighboring point; (**c**) candidate templates of different sizes; (**d**) entropy curve of the pattern histogram; (**e**) the optimal template size specified by the elbow-point detection.

Figure 5 provides a conceptual example to explain how to extract the morphological characteristics from the TI. Prior to the clustering step, the computer extracts every pattern with a template. Next, hierarchical clustering is performed to analyze the patterns. As the initial condition, each instance is regarded as an individual group. Subsequently, the computer combines the two groups with the shortest distances. It is worth noting that the distance between two pattern groups is a key point in hierarchical clustering. To prevent the influence of outliers, we select the average linkage. In other words, the mean distance between all members in the two groups is the similarity metric. The program successively performs the merging function until a stopping condition is met. In this simplified case, we specify the distance threshold as 0.34. Namely, our program does not combine groups whose distance is larger than 0.34. Thus, five pattern groups are detected in this case. Based on the clustering results, the program records the number of members in each pattern group. As displayed in Figure 5f, a pattern distribution is generated as an indicator of the TI.

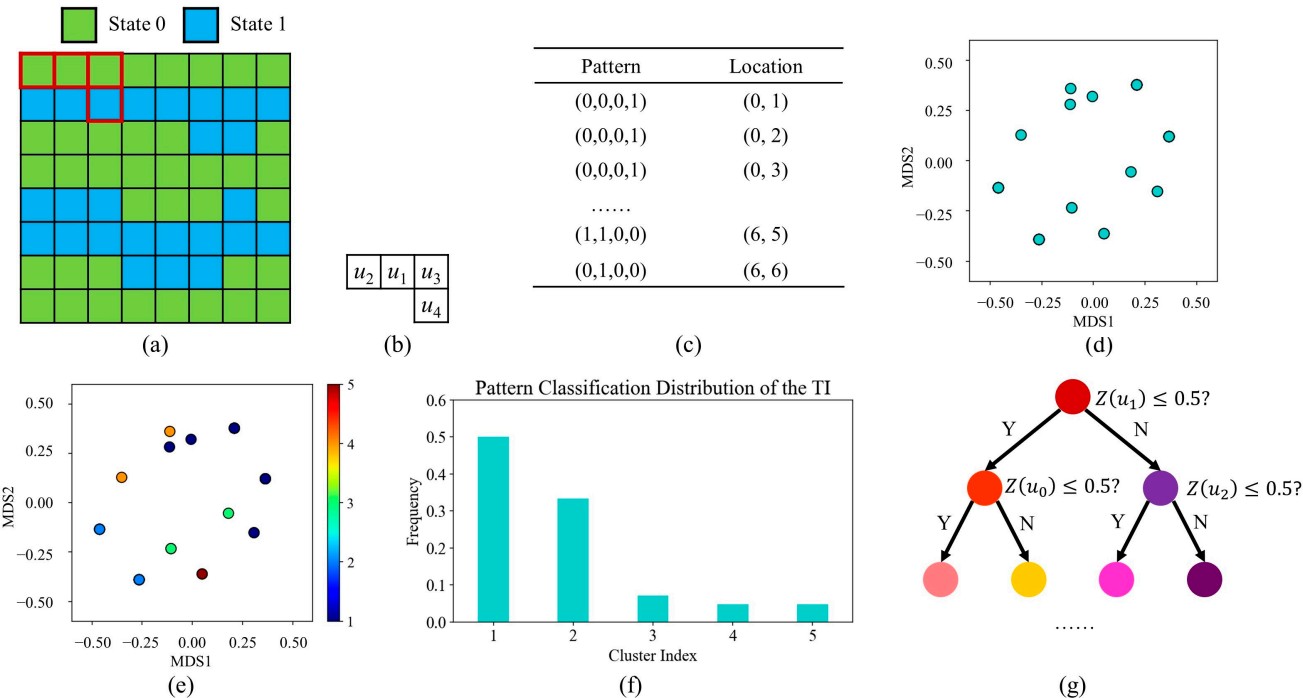

**Figure 5.** The hierarchical clustering of training patterns. (**a**) Training image and the first spatial pattern; (**b**) the irregular template designed by the correlation-driven method; (**c**) pattern dataset; (**d**) patterns in MDS feature space; (**e**) hierarchical clustering result in MDS feature space; (**f**) pattern-classification distribution of the TI; (**g**) decision-tree classifier trained by the clustering result.

After the clustering procedure, each pattern in the TI has a label. Next, our program focuses on characterizing the geological model in accordance with the clustering results. Similar to ANODI, a classification program is launched to create a pattern distribution from the realizations. With the objective of improving the performance, there are two important developments. First, the proposed method employs a decision tree [35] as the classifier. Compared with the nearest-neighbor approach in ANODI, the benefit of the decision tree lies in the efficiency to complete classification tasks. As an eager learning technique, the decision tree attempts to extract valuable information from the training data. Based on a flowchart-like structure, the prediction is achieved by continuously examining an attribute of the query instance. The time complexity is heavily dependent on the number of features. In comparison, the nearest-neighbor classifier is a lazy learner. The interpretation of the training data is generally delayed until the computer inputs a query. Given the large amount of patterns, the distance computation with every representative pattern leads to considerable time consumption.

The second improvement in our classification method is that every pattern in the TI is considered during the classifier training step. There is no pattern-selection step in the proposed program. It is noticeable that the number of training examples does not have a substantial influence on the speed of the decision tree classifier. In comparison, ANODI identifies the pattern that is closest to the group centroid as a representative. To save running time, the rest of the observations are removed by the classification program. The geological model characterization program suffers from information loss.

A simplified example with which to discuss the geological model characterization is shown in Figure 6. It can be seen that the first realization has a similar structure to the TI in Figure 5a. Two channels flow from the left to the right side. In contrast, the $RE_2$ exhibits different behavior. There are no connected components between the three blue areas. Our program contains four procedures to distinguish two realizations. (1) The computer extracts geological patterns with a template. As shown in Figure 6b, a dataset is created to store all the patterns. (2) Each pattern is classified with the decision tree. In

this case, we allocate the geological patterns into five groups. (3) The program records the size of each pattern group. As shown in Figure 6c,f, a pattern classification distribution is independently generated. (4) Based on Equation (1), Jensen–Shannon divergence is applied to measure the similarity. A large value for distance indicates that there is a huge difference between two geological models. In this case, the divergence between the *TI* and the first realization is $dis_{JS}(TI, RE_1) = 0.11$. By comparison, $dis_{JS}(TI, RE_2) = 0.24$ is yielded when the computer focuses on the *TI* and the second realization. These computation results are consistent with the morphological characteristics of the three models.

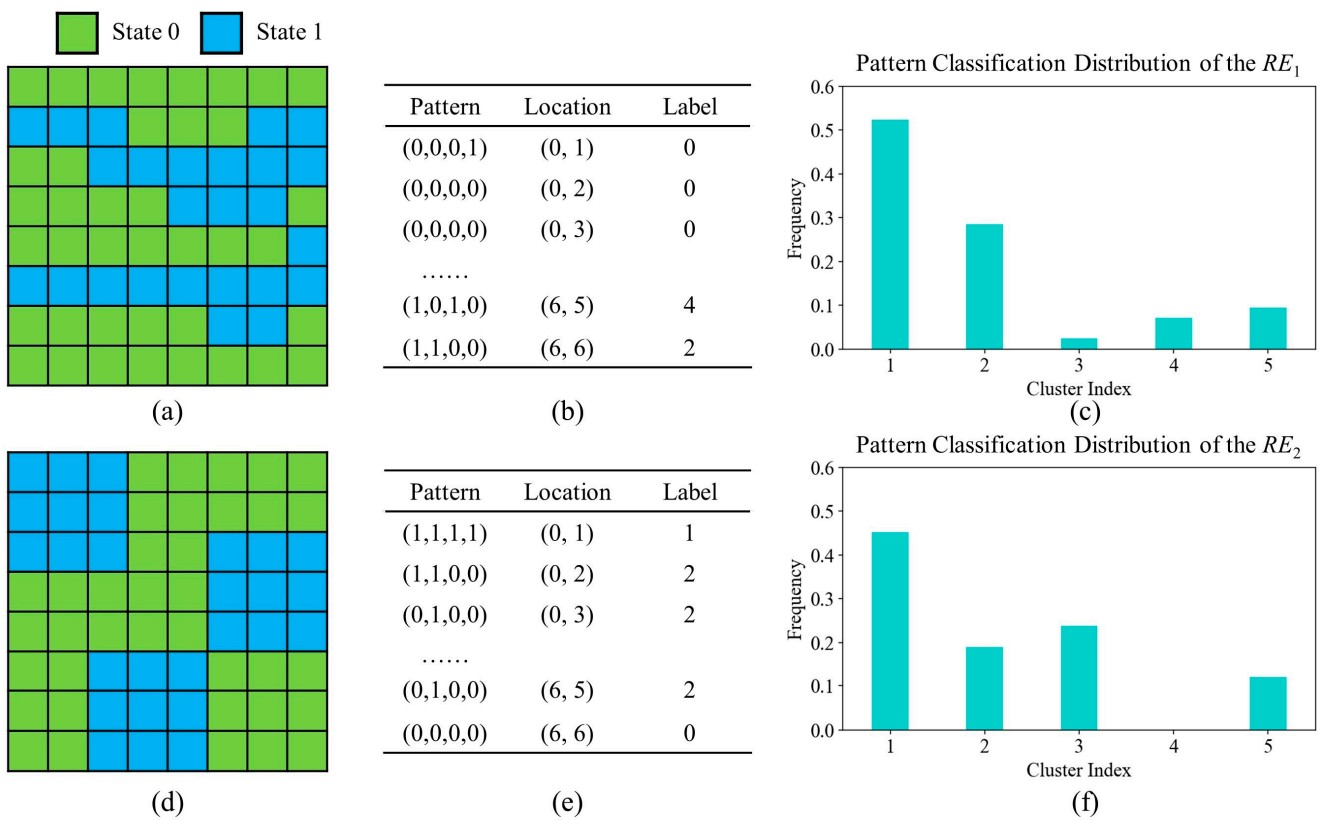

**Figure 6.** Geological model characterization with the decision-tree classifier. (**a**) The first realization $RE_1$ with two channels; (**b**) dataset pattern constructed by $RE_1$; (**c**) pattern classification distribution of $RE_1$; (**d**) the second realization $RE_2$ with three isolated patches; (**e**) dataset pattern constructed by $RE_2$; (**f**) pattern classification distribution of $RE_2$.

Based on the JS divergence, our PCD method is able to evaluate the geostatistical variability and rank modeling methods. As with MPH and ANODI, the average distance between the TI and the realizations becomes a descriptor of the pattern reproduction ability. On the other hand, the spatial uncertainty is quantified by the average distance between geological realizations. In addition, we insisted on applying the ratio to compare the two simulation methods. According to Equations (4)–(6), three ratios were output as the comparative results.

### 3.3. Automatic Resolution Importance Assignment with a Stacking Strategy

On the basis of the template design and model characterization procedures described above, our program has the ability to measure geometrical similarity. One technical limitation is that the template used above can only observe short-radius structures. As noted in Section 2.2, ANODI applies a multi-resolution pyramid to analyze small-scale as well as long-range structures. According to Equations (7) and (8), the morphological similarity is combined across different resolutions. However, the geostatistical evaluation program suffers from the fixed weights in the aggregation formula. The underlying assumption

behind the current weight assignment is that the long-scale structure is less important than the fine-grain pattern in the geostatistical modeling. However, this assumption is not always true. For example, the long-range structure is a contributing factor in water resource management. With the aim of creating high-quality realizations, MPS is dedicated to reproducing the long-distance connectivity in flume systems and subsurface aquifer systems [10,17]. In petroleum engineering, the permeability of reservoir rocks heavily relies on the pore space [36]. Large-scale connected components play an important role in geostatistical modeling.

With the objective of developing the multi-grid strategy and improving the evaluation accuracy, a stacking framework was proposed to automatically assign the importance of each grid. The motivation for this proposal was the ensemble-learning method in the field of machine learning [35]. As a meta-learning framework, the stacking method takes advantage of two or more base machine-learning programs. There are two basic steps. First, a collection of base machine-learning programs are trained based on the same dataset. In general, it is helpful to use a diverse range of learning techniques. Second, the computer applies a meta-estimator to assess the effectiveness of each base program. The meta-estimator focuses on exploring the relationship between the predictions made by the base learner and the ground-truth labels.

Figure 7 provides an example to illustrate the stacking framework in our PCD. In this case, we specify $G = 2$ due to the limited size of the TI. There are two kinds of classifier in the proposed method. On one hand, the base classifier focuses on identifying the relationship between the template center and neighboring points. Multiple-point statistics information is effectively captured. For example, the short-range structure in the TI is analyzed by the first base classifier. By comparison, base classifier 2 attempts to capture long-range patterns with an extending template. On the other hand, the computer applies a meta-classifier to evaluate the strength of each base classifier. For a TI with long-term connectivity, there is a strong correlation between the center point and the conditioning points gathered by a large template. Base classifier 2 plays an influential role in the point prediction. A bigger contribution is assigned to the large-scale grid.

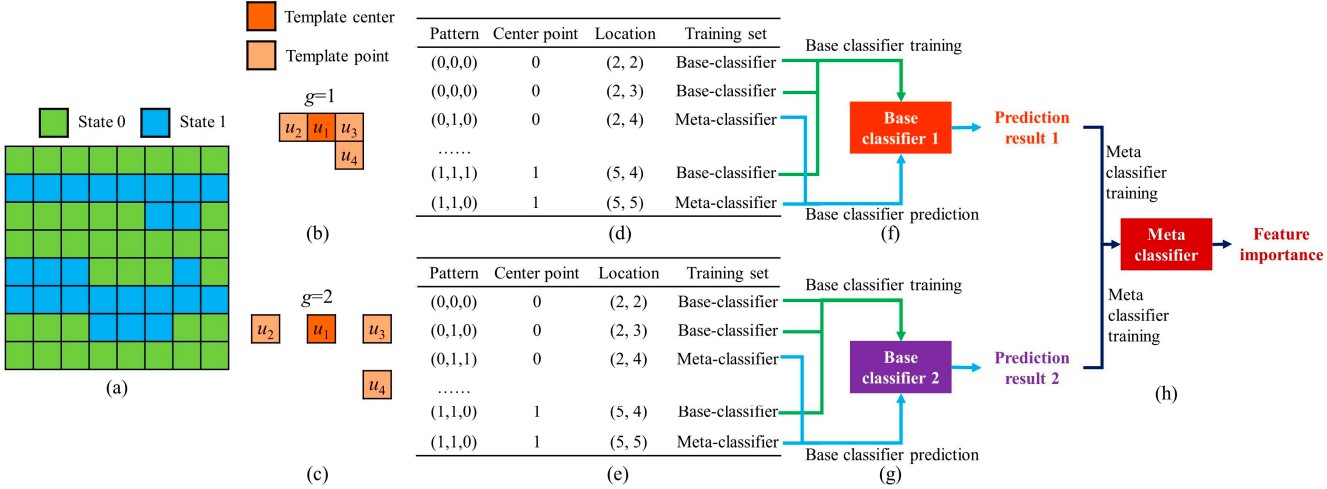

**Figure 7.** A stacking framework to compute the importance of each grid. (**a**) Training image; (**b**) a compact template with four points; (**c**) a sparse template with an extending radius; (**d**) patterns captured by the compact template; (**e**) patterns captured by the extending template; (**f**) the first base classifier. The green lines represent the instances used to train the classifier. By comparison, the patterns that are used to make point predictions are highlighted in blue; (**g**) the second base classifier; (**h**) the meta-classifier, which outputs the importance of each grid.

Furthermore, our program applies two key modifications to the traditional stacking framework. First, the base classifier is trained with different data. Based on the template

with varying receptive fields, the computer captures spatial patterns across different resolutions. The multi-grid features are individually input to each base classifier. Second, the base classifiers share the same classification technique. In our program, the decision tree is applied. By comparison, the stacking program previously used in the machine-learning community encourages the utilization of heterogeneous classification techniques.

The detailed steps in the proposed method are as follows. (1) Viewing the irregular template as the prototype, our program creates a set of expanding templates to extract patterns from different grids. (2) A dataset is generated to collect training patterns. A noticeable phenomenon is that our program individually stores the center point and the template points. The conditioning points gathered by the template are the input features in the classification task. By comparison, the center point is viewed as the target variable. (3) The computer divides each pattern dataset into two subsets. Our program randomly assigns 70% instances into the first set. Accordingly, 11 patterns are selected in this case. The five remaining patterns comprise the second subset. (4) Two base classifiers are trained by the first pattern subset. The proposed method applies the decision tree to build the bridge between the template center and the neighboring points. After the training step, the patterns in the second subset are fed into these two base classifiers. Thus, five predictions are separately produced. (5) Our program trains the meta-classifier. In this framework, random forest is employed as the meta-classifier. The meta-classifier focuses on determining the relationship between the base-classifier outputs and the center points. (6) The meta-classifier outputs the feature importance as the contribution of each base classifier. In this case, the importance values of the two resolutions are 0.51 and 0.49, respectively. These computational results imply that the role of long-range structures is close to that of the small-range patterns. In other words, reproducing large-scale connectivity is an important aspect in this modeling scenario. However, the current MPH and ANODI cannot automatically control the weight assignment according to the simulation scenario. The significance of long-range structures is underestimated in this conceptual case.

## 4. Applications

### 4.1. A 2D Benchmark Channel Model with Anisotropic Structures

As the first application, the benchmark channel model was employed to examine the proposed PCD. Based on the TI shown in Figure 8a, we independently launched a group of MPS programs to generate 200 realizations. At first, single normal equation simulation (SNESIM) [18] and filter-based simulation (FILTERSIM) [15] were individually performed by Stanford Geostatistics Modeling Software (SGeMS) [37]. We applied a template with a size of $9 \times 9$ and a multigrid strategy of $G = 3$ in SNESIM. By comparison, the default setting was utilized by FILTERSIM. The sizes of the searching template and the pasting template were specified as $11 \times 11$ and $7 \times 7$, respectively. Next, we implemented three database-based MPS programs. Improved parallelization (IMPALA) [38], column-oriented simulation (CSSIM) [39], and nearest-neighbor simulation (NNSIM) [10] were performed to create the channel models. The parameters in IMPALA and CSSIM were the same as those in SNESIM. In NNSIM, our program specifies the cosine distance threshold and the number of teachers as 0.1 and 5, respectively. Finally, direct sampling (DS) [40] and tree-based direct sampling (TDS) [17] were carried out in this stochastic simulation scenario. According to the experiments conducted by Meerschman et al. [41], three predefined parameters of DS are $N^{DS} = 30$, $t^{DS} = 0.05$ and $f^{DS} = 0.5$. In addition, TDS is activated by a clustering tree with a height of 9. The first realization created by these methods is shown in Figure 8.

Based on the geological models discussed above, our PCD was performed to quantitatively assess the MPS simulation quality. As the first step, our program generated a template according to the intrinsic characteristics of TI. The computation results are shown in Figure 9. It is clear that there was strong anisotropy in the channel image. The spatial correlation and connectivity in the horizontal direction were more intensive than in the vertical direction. In order to conserve the channel structures, the proposed template design method sequentially collects points with strong correlations. The pattern entropy function

and the elbow point detection technique were employed to determine the template size. As displayed in Figure 9d, 26 points were involved in our irregular template.

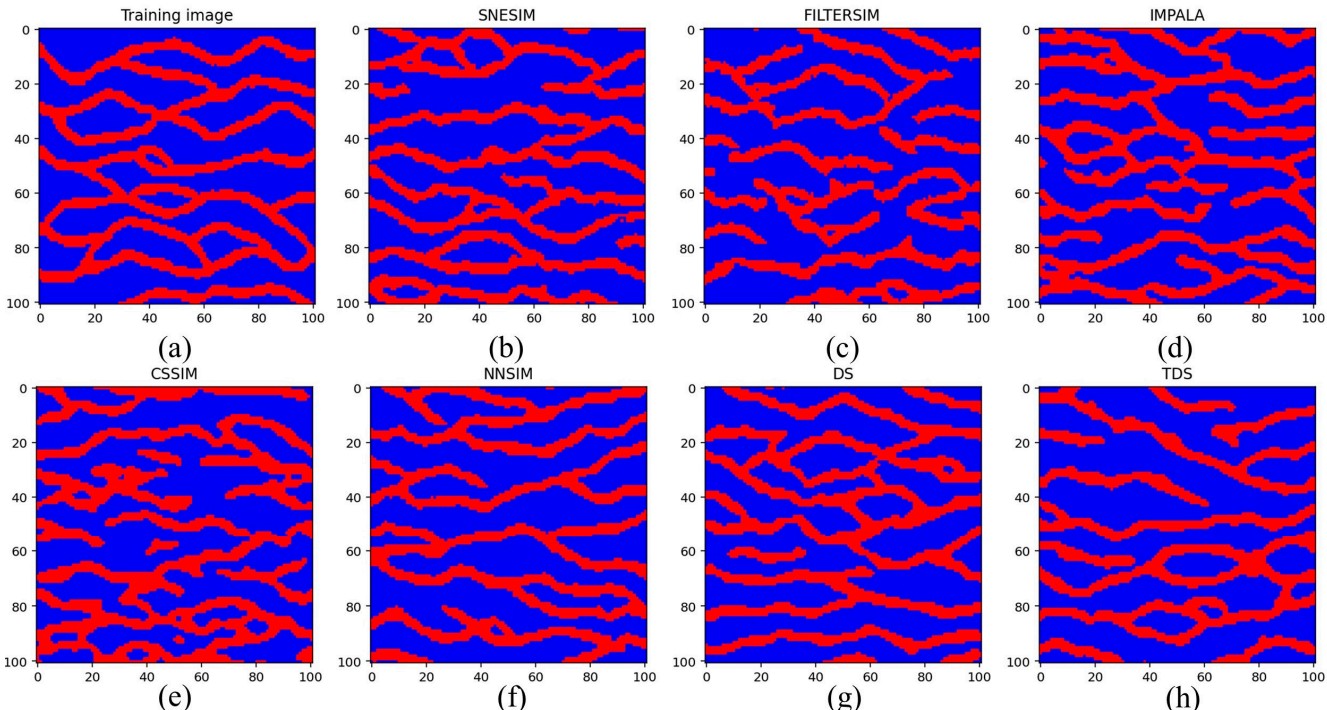

**Figure 8.** Channel realizations created by MPS programs. (**a**) Training image; (**b**) SNESIM model; (**c**) FILTERSIM model; (**d**) IMPALA model; (**e**) CSSIM model; (**f**) NNSIM model; (**g**) DS model; and (**h**) TDS model.

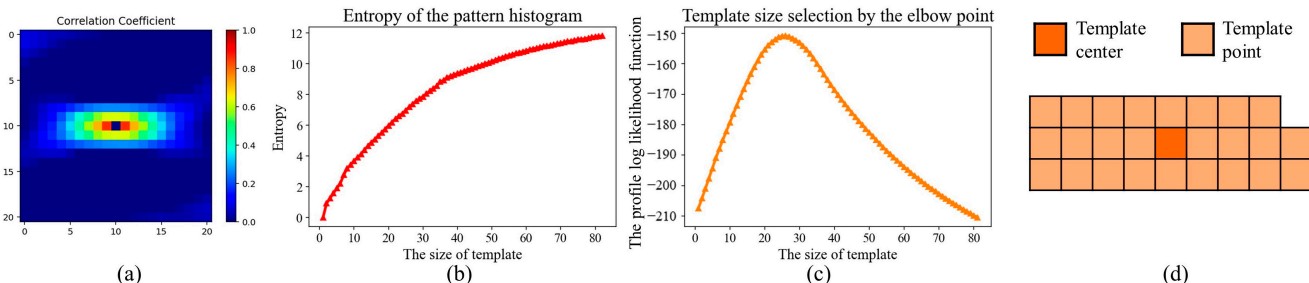

**Figure 9.** The correlation-driven template design in the channel simulation. (**a**) The correlation coefficient of each template point; (**b**) entropy curve of the pattern histogram; (**c**) template size selection by the elbow point detection. (**d**) the optimal template with 26 points.

Next, the stacking framework was launched to assign the resolution importance. In this case, we specified the number of $G = 4$. As the base classifiers, four decision trees focused on predicting the state of the template center according to the neighboring points across different resolutions. Moreover, the random forest technique was used as the meta-classifier to validate each base classifier. Consequently, the resulting weight vector was [0.56, 0.24, 0.17, 0.03].

Next, the hierarchical clustering method was used to organize the training patterns. For instance, our program extracted 9207 patterns from the finest grid $g = 1$. Based on the hierarchical clustering program associated with a distance threshold of 0.1, 339 pattern groups were found. Subsequently, our program performed a decision tree to characterize the MPS realizations. The frequency of each category became a descriptor of the morphological characteristics. Finally, the JS divergence and the weighted aggregation technique were carried out to distinguish the two geological models. Based on Equations (7) and (8),

we quantified the pattern reproduction and spatial uncertainty by the average distance. The relative behavior of each MPS program was compared by Equations (4)–(6). In this case, we applied the SNESIM realizations as the method *B*. The computation results are highlighted in Figure 10a. Furthermore, we activated multi-dimensional scaling (MDS) to visualize the calculation results. In the feature space, each node represents a geological model. The two close points imply that there was intensive compatibility between two MPS realizations. Figure 11a displays the MDS visualization results. In order to avoid visual confusion, we partitioned the point cloud into three parts. First, the SNESIM, IMPALA, and CSSIM realizations are emphasized. Second, the blue points display the dispersals of NNSIM, DS, and TDS models. Third, the FILTERSIM realizations are presented in yellow.

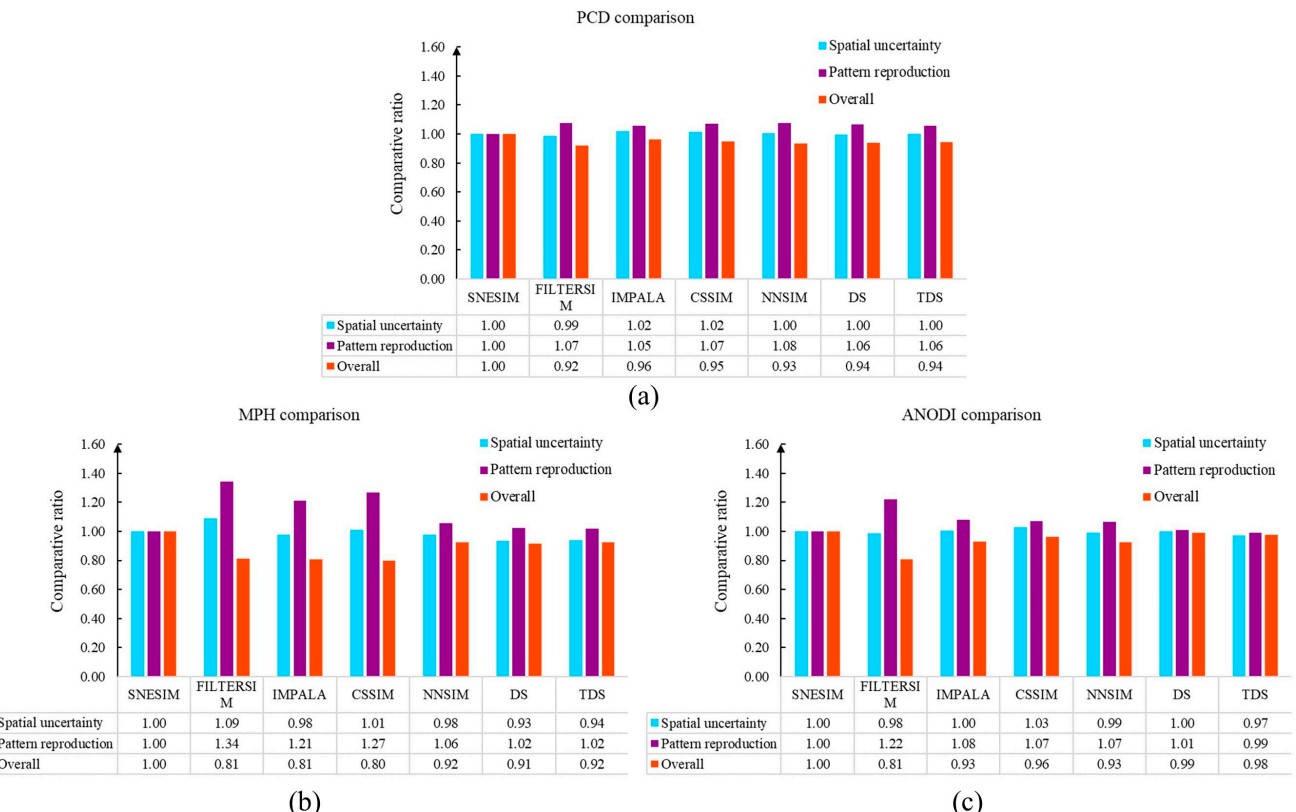

**Figure 10.** Geostatistical-quality-evaluation results from (**a**) PCD; (**b**) MPH; and (**c**) ANODI.

With the aim of performing an extensive comparison, we implemented MPH and ANODI in this case. In MPH, a template of size $3 \times 3$ was applied to extract the patterns. Therefore, there were 512 possible values in the multiple-point histogram. Furthermore, we employed a template of size $7 \times 7$ and a multi-grid strategy with $G = 4$ in ANODI. The number of pattern clusters was specified as 40. The computation results of MPH and ANODI are shown in Figures 10 and 11.

Two key observations were made based on the PCD results. (1) According to the comparative ratios, SNESIM exhibited a competitive performance. In Figure 10a, all the overall ratios are lower than 1.0. The main reason is that the postprocessing step in SNESIM plays a positive role in improving simulation quality. Mismatching structures are upgraded by the re-simulation step. (2) Based on Figure 11a, the preceding MPS programs can be partitioned into three groups. First, there was a strong similarity between the SNESIM, IMPALA, and CSSIM realizations. The orange, green, and turquoise points are located at the top. Next, the NNSIM, DS, and TDS models had relatively small distances. It is clear that the blue points constitute a group in the bottom-left. Finally, the FILTERSIM realizations in yellow were in disagreement with the other methods. A similar phenomenon can be observed in Figure 11b. However, ANODI did not highlight a significant difference between the MPS

realizations. The main reason for this finding lies in the pattern-matching mechanism in the MPS framework. SNESIM, IMPALA, and CSSIM employ the pruning strategy to find a compatible instance. When a completely matching pattern does not exist, the program discards the conditioning point with the maximum distance. By comparison, distance computation is an essential component in NNSIM, DS, and TDS. These three programs apply Hamming distance to distinguish between patterns. Furthermore, the core idea in FILTERSIM is to utilize a set of filters to organize training patterns. The program classifies 2D patterns based on the convolution scores with six predefined kernels.

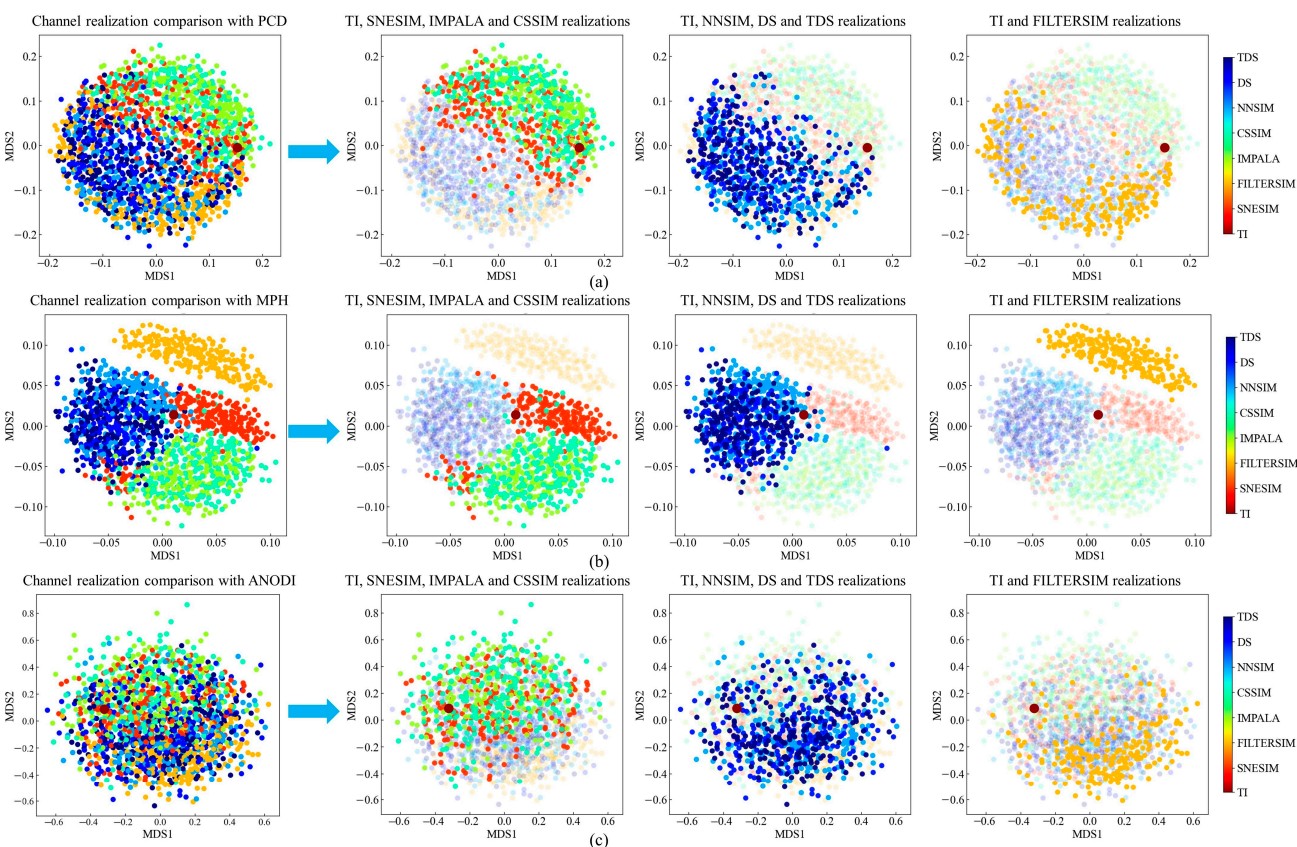

**Figure 11.** Uncertainty-quantification results based on (**a**) PCD; (**b**) MPH; and (**c**) ANODI.

With the purpose of improving practicability, we investigated the parameter sensitivity of PCD, MPH, and ANODI. SNESIM and DS models were adopted as method *A* and method *B*, respectively. On one hand, we studied the influence of the distance threshold in the PCD. The comparison results are shown in Figure 12a. As the only user-defined parameter, the distance threshold in the hierarchical clustering had a small effect on the evaluation result. Three ratios did not demonstrate intensive variation. On the other hand, parameter setting is a key module within MPH and ANODI. There were strong fluctuations in the comparative ratio curves. The changing behavior creates difficulties in the quantitative evaluation of the MPS modeling quality.

## 4.2. A 2D Non-Stationary Flume System with Morphologically Complex Structures

Autogenic variability is a fundamental aspect of numerous Earth surface systems. Schedit et al. [1] and Hoffimann et al. [3] simulated a delta evolution in laboratory experiments. Based on the overhead snapshots, a group of flume realizations were created to express spatiotemporal uncertainty in a channelized transport system. In this section, we focus on examining the performance of PCD in the context of multiple geological categories.

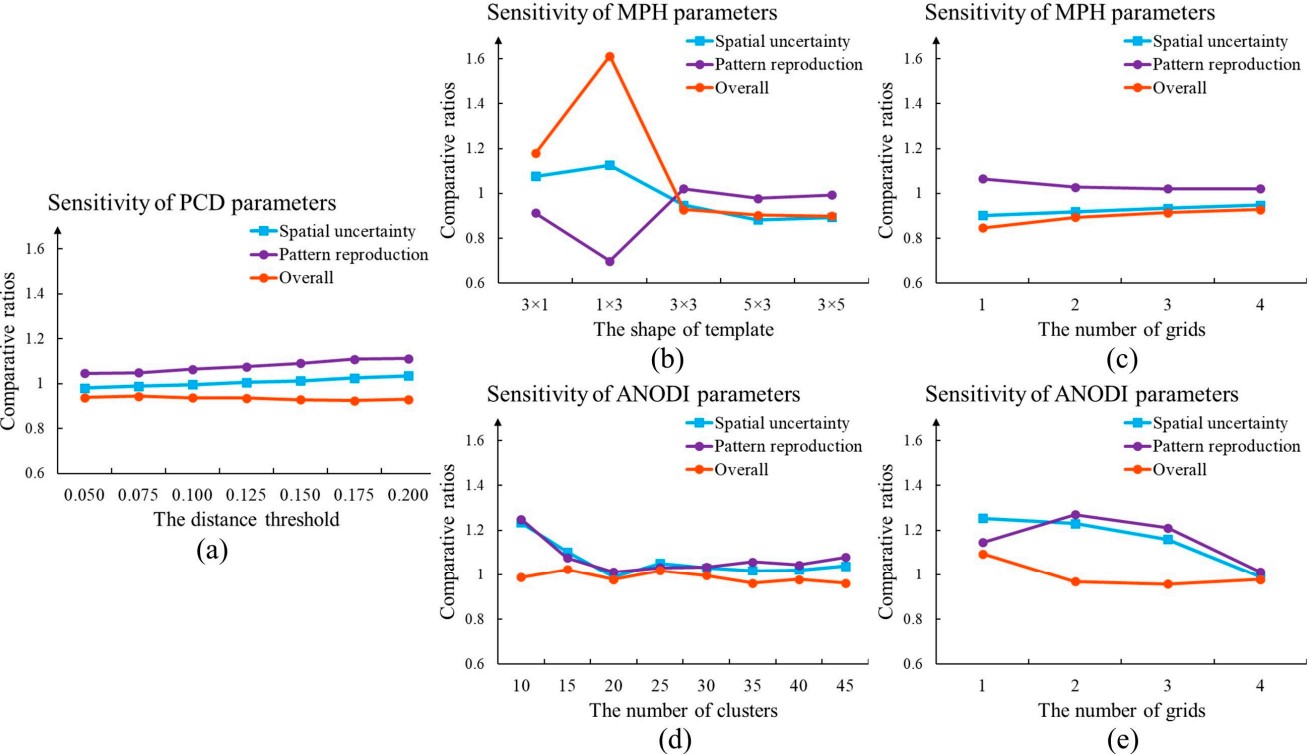

**Figure 12.** Parameter sensitivity of three evaluation methods. (**a**) The influence of distance threshold within PCD; (**b**) the influence of template within MPH; (**c**) the influence of the grid number within MPH; (**d**) the influence of the cluster number within ANODI; (**e**) the influence of the grid number within ANODI.

Figure 13a exhibits three TIs. The sediment is expressed by the blue area, while the intensity of flow is reflected by other colors. In the first TI, there are only two geological categories. By contrast, three and four states are presented in the second and third TIs, respectively. We independently implemented NNSIM and TDS to generate the non-stationary flume model. 10 flume realizations were individually created by two programs. With the aim of addressing non-stationarity, the computer introduces the auxiliary variable into the MPS framework. Based on the proximity to the original point, the simulation grid is split into four subareas. Therefore, NNSIM and TDS create an independent pattern dataset for each area. With the purpose of improving the simulation quality, NNSIM applies a template of size $9 \times 9$ and a multi-grid strategy of grid 4. We specified the parameters in TDS as $N^{DS} = 30$, $t^{DS} = 0.05$ and $f^{DS} = 0.5$. Moreover, the height of the clustering tree was configured as 9. Figure 13c,d provide the first realization of the methods described above. Detailed explanations of the TI and MPS simulation are provided in [1,10].

The proposed PCD was applied to rank the MPS programs. TDS and NNSIM were specified as methods *A* and *B*, respectively. In order to address geometrically complex structures, we specified the distance threshold in the hierarchical clustering as 0.15. Within the multi-grid framework, the number of grids was set as 4. The comparative ratios and MDS visualization are shown in Figure 14. The overall ratios in the three scenarios were close to 1.00. In addition, the point clouds had comparable dispersal. These findings suggest that NNSIM and TDS had similar accuracy in this simulation task.

We carefully investigated the parameter setting issue. One key advantage of the proposed method is that PCD applies the correlation-driven template, a combination of hierarchical clustering and decision tree, and a stacking framework. A set of parameters are automatically computed according to the morphological characteristics of the TI. As shown in Figure 15, the distance threshold did not play an influential role in the evaluation results. The adaptive parameter configuration within PCD is beneficial to quantitatively

assess the progress of each MPS program. By contrast, ANODI is heavily dependent on the parameter specification. In order to extract complex patterns, ANODI applies a template with a size of $7 \times 7$. The effects of the number of clusters and the number of grids were checked. As displayed in Figure 16, there were remarkable differences between the ANODI outputs. It is challenging to objectively compare the strengths of NNSIM and TDS.

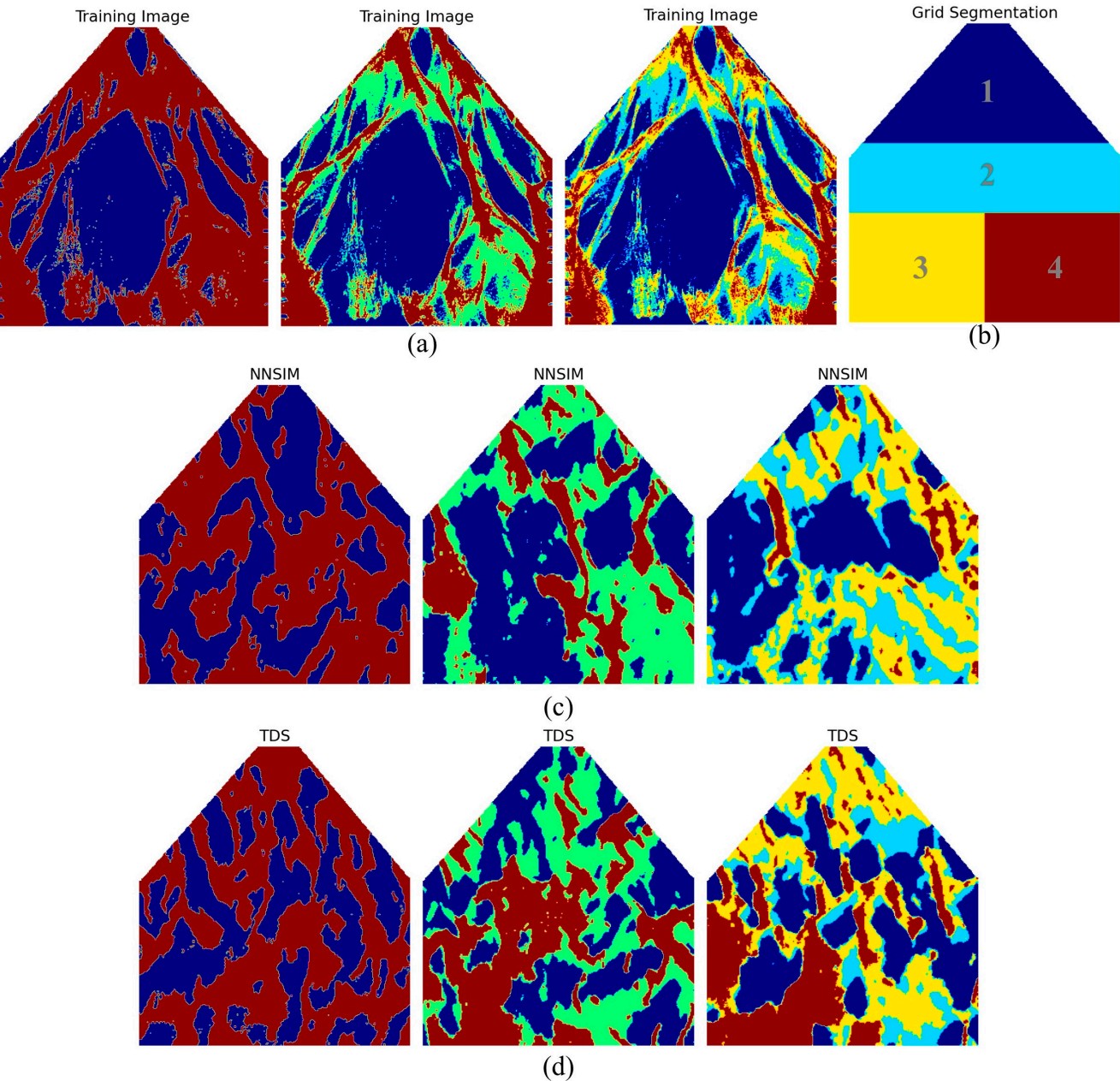

**Figure 13.** MPS realizations in the flume modeling. (**a**) Three training images with multiple geological categories; (**b**) auxiliary variable in MPS simulation. The numbers indicate the indices of four subareas; (**c**) NNSIM realizations in three modeling scenarios; (**d**) TDS realizations in three modeling scenarios.

Moreover, we investigated the sensitivity of the control parameters in relation to the NNSIM simulation quality. On one hand, the template is an essential component in the collection of conditioning points. An expanding template has a positive effect on the reproduction of complicated structures. NNSIM applies templates with sizes of $3 \times 3$, $5 \times 5$, $7 \times 7$, $9 \times 9$, $11 \times 11$, and $13 \times 13$. Furthermore, $G = 3$ is utilized to extract patterns across different scales. On the other hand, the multi-grid strategy provides a

valuable tool to simulate patterns across different resolutions. The numbers of grids were configured as 1, 2, 3, and 4, respectively. A template with a size of 9 × 9 was employed. For each parameter specification, 20 flume models were independently generated by NNSIM.

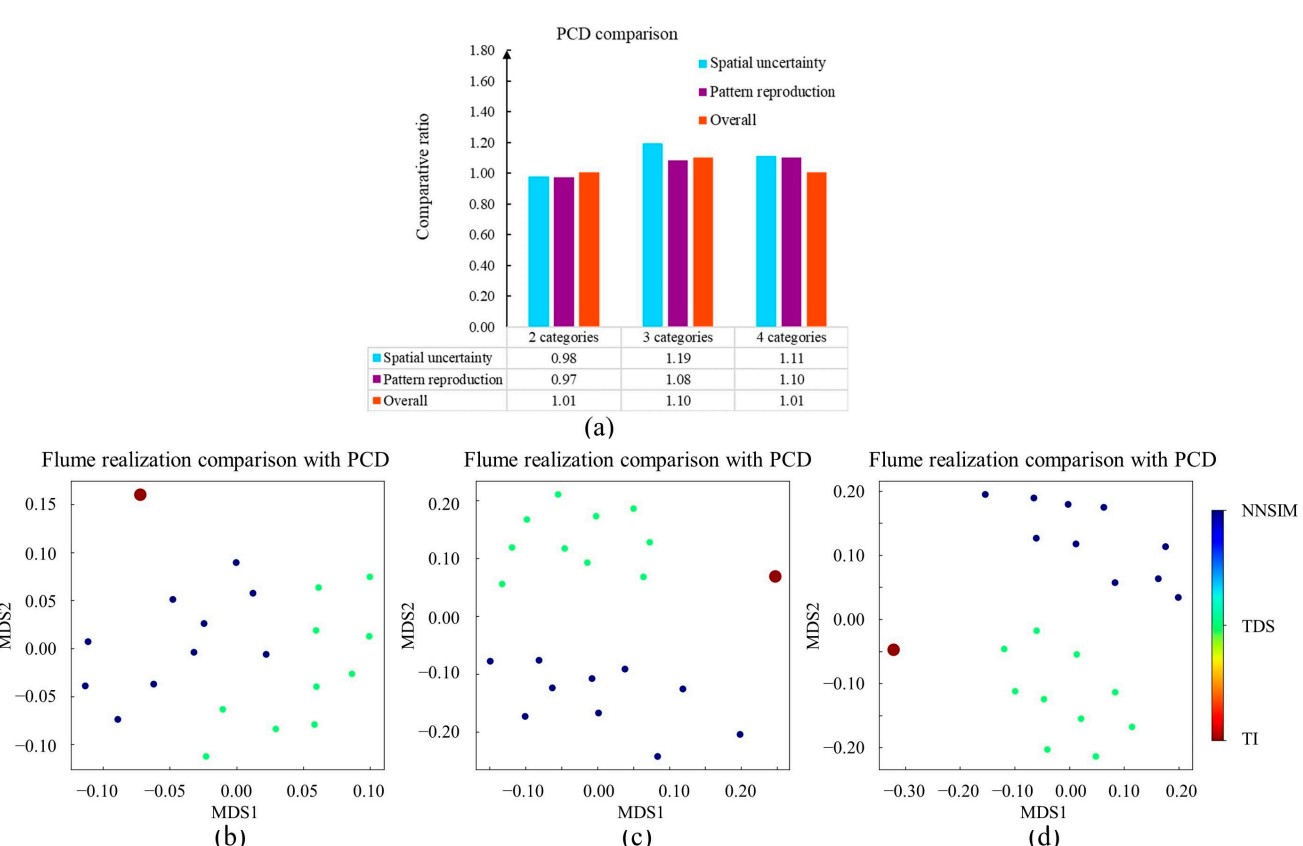

**Figure 14.** PCD calculation results. (**a**) Comparative ratios between NNSIM and TDS; (**b**) uncertainty quantification in the two-facies flume simulation; (**c**) uncertainty quantification in the three-facies flume simulation; (**d**) uncertainty quantification in the four-facies flume simulation.

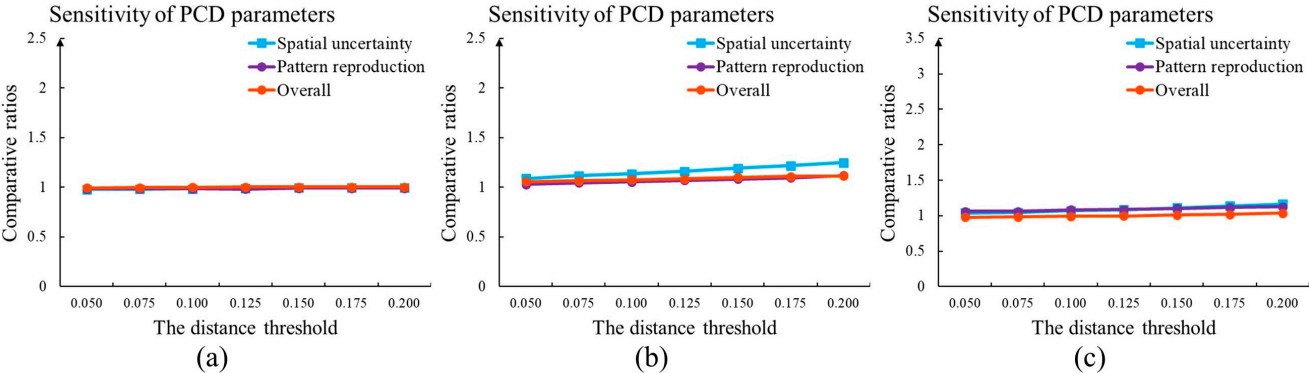

**Figure 15.** Parameter sensitivity of PCD in (**a**) two-facies flume simulation; (**b**) three-facies flume simulation; (**c**) four-facies flume simulation.

We applied PCD to find reliable realizations and quantify the uncertainty. The model set produced by the first parameter setting was defined as method *B*. The evaluation results are shown in Figure 17. The findings were as follows. (1) MPS simulation benefits from an extending template. In Figure 17a–c, the purple columns have decreased with the development of the template size. This reveals that MPS methods are able to effectively reproduce patterns in the simulation domain. Furthermore, the expanding template contributed to the increase in the overall ratios, which are expressed in orange. (2) The multi-grid strategy

is a key module to improve MPS quality. According to Figure 17d–f, a high grid value not only enriches the spatial uncertainty but also reduces the differences between the TI and the realizations.

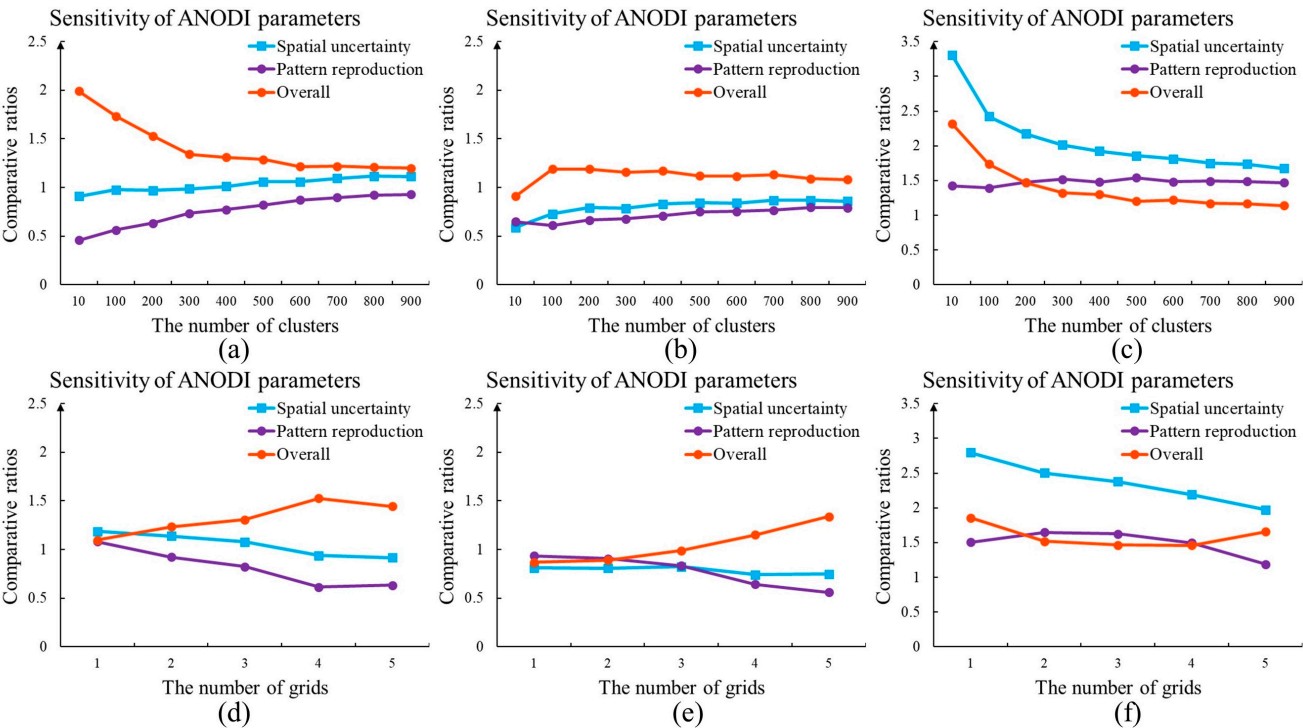

**Figure 16.** Parameter sensitivity of ANODI in the three modeling scenarios. (**a**) The influence of cluster number on two-facies simulation; (**b**) the influence of cluster number on three-facies simulation; (**c**) the influence of cluster number on four-facies simulation; (**d**) the influence of grid number on two-facies simulation; (**e**) the influence of grid number on three-facies simulation; (**f**) the influence of grid number on four-facies simulation.

In order to complete an extensive comparison, we inspected more parameter combinations within NNSIM. The influences of the template size and the multi-grids were simultaneously investigated. In this case, our PCD selected the realizations created by a template $9 \times 9$ and two grids as method $B$. The computational results are shown in Figure 18. First, the red color in the spatial uncertainty map indicates that the model sets demonstrated a high level of diversity. Second, the close proximity to the TI is emphasized by the white and blue at the second column of Figure 18. Third, the competitive programs are highlighted by the red in the overall ratio matrix. Apparently, the use of a large template and a multi-grid approach are helpful to create realistic realizations. The program provides its best performance when a template of size $9 \times 9$ and three grids are employed. In addition, it should be noted that there are blue squares in the bottom-right area at the last column in Figure 18. This implies that NNSIM does not provide reliable models when a template of size $13 \times 13$ and $G = 4$ is utilized. The main reason for this is that the large template and grids contain numerous conditioning points in MPS simulations. The uncorrelated points not only provide redundant information but also create computational burden. Therefore, the parameter selection is a key component within the MPS framework. An unsuitable extension in template size and grids prevents the MPS program from outputting favorable models.

### 4.3. A 2D Subglacial-Bedrock-Elevation Model with Continuous Variable

Knowledge of the topography beneath Antarctica and Greenland ice sheets is essential for a wide range of glaciological investigations. With the aim of better predicting subglacial

flow behavior, it is necessary to create a collection of high-resolution topographic models and express spatial uncertainty. In particular, the characteristics of the subglacial topography of the Thwaites Glacier in the Amundsen Sea Embayment have received considerable attention [42]. The accelerating ice loss in Thwaites Glacier has a substantial influence on the stability of the West Antarctic Ice Sheet [43]. Based on the non-stationary multiple-point geostatistics method, Yin et al. generated a set of realistic topographic models [7]. On one hand, the stochastic modeling method is guided by 166 high-quality topographic training images, which are extensively sampled from the deglaciated regions in Arctic and Antarctica. On the other hand, ice penetrating radar data become the hard data in this simulation task.

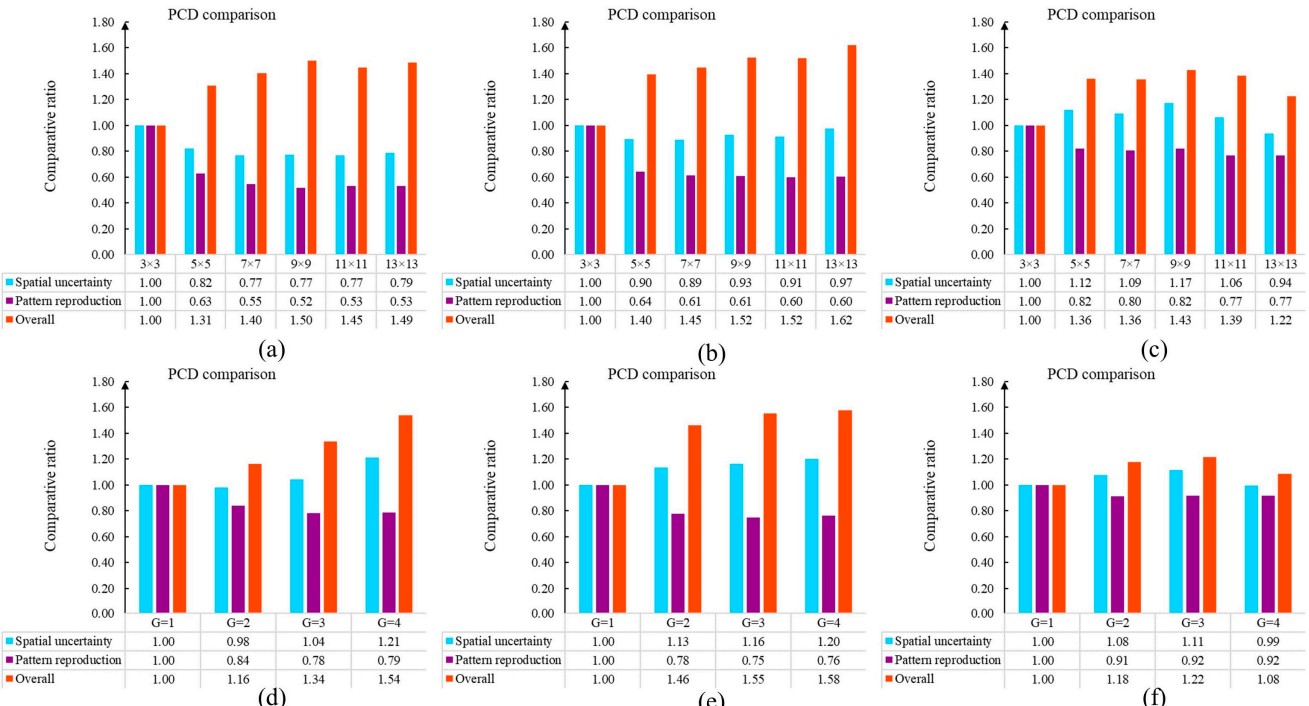

**Figure 17.** PCD evaluation to check the parameter sensitivity within NNSIM. (**a**) PCD comparison between different templates in two-facies simulation; (**b**) PCD comparison between different templates in three-facies simulation; (**c**) PCD comparison between different templates in four-facies simulation; (**d**) PCD comparison between different grids in two-facies simulation; (**e**) PCD comparison between different grids in three-facies simulation; (**f**) PCD comparison between different grids in four-facies simulation.

Aiming at generating diverse models, three geostatistical simulation methods were applied in this case. (1) We used the Kriging method to generate a subglacial topographic model according to the radar data. As a deterministic method, Kriging produces only one realization. (2) Sequential Gaussian Simulation (SGSIM) was carried out to create 10 stochastic realizations. The program estimates the variogram on the basis of radar lines. (3) We implemented non-stationary multiple-point geostatistics to create digital elevation models. There were two major steps in this process. First, a training-image-transition method was used to find the optimal prior model for each local subarea. Second, the computer activated direct sampling (DS) to complete the gap-filling task. In this case, MPS generated 10 subglacial topographic models. The first realization of three geostatistical modeling programs is shown in Figure 19. The authors of [7] discuss the technical details of previous methods.

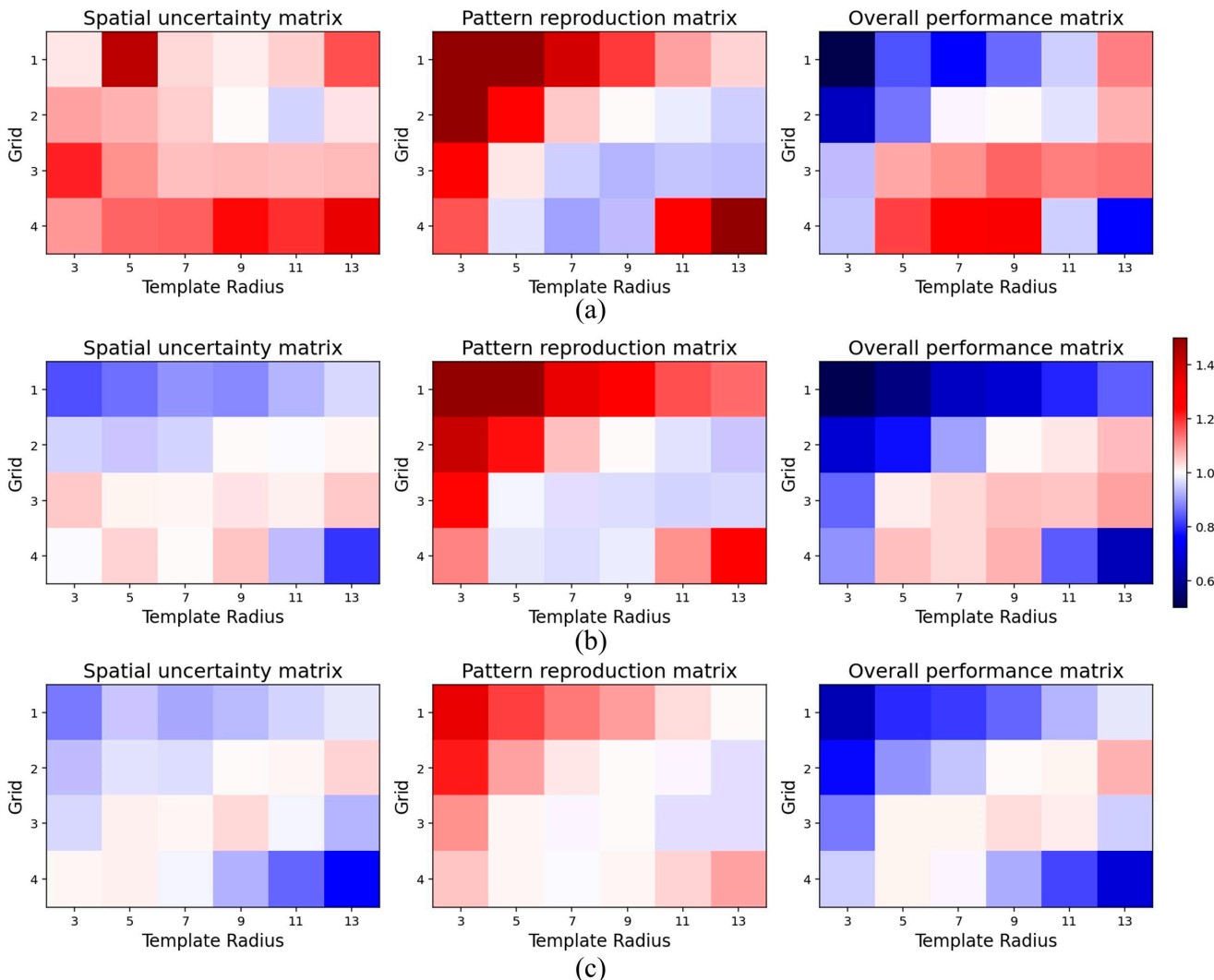

**Figure 18.** PCD evaluation of various NNSIM parameter combinations. (**a**) Two-facies simulation; (**b**) three-facies simulation; (**c**) four-facies simulation.

In this section, the proposed PCD is examined by a large-scale stochastic simulation with continuous variable. A noticeable phenomenon is that there was no training image in this evaluation task. Kriging and SGSIM were used to explore the spatial dependency in accordance with the radar lines. In contrast, the non-stationary MPS featured a min–max normalization on 166 TIs. The standardization of the bedrock elevation helped MPS concentrate on reproducing the morphological structures. Accordingly, it was not reasonable to directly compare the 166 TIs and geostatistical realizations in this case.

With the aim of mitigating the absence of TI, our PCD was used to analyze the spatial patterns in the first MPS realization. As the first step, the correlation-driven template design method was launched. To calculate the entropy curve, we applied the multi-level thresholding program to tackle the continuous variable. The bedrock elevation was uniformly partitioned into several bins. Therefore, the number of geological categories was an important parameter. Figure 20b provides the segmentation results. As shown in Figure 20c, our program computed the template size according to the intrinsic characteristics of the categorical models. There were two findings. (1) The template size ranged from 18 to 22 when the number of facies was lower than 6. (2) With the development of the geological categories, the template size progressively decreased. The main reason is that there was an exponential relationship between the number of geological states and the amount of possible pattern configurations. A high value of geological categories leads to a sparse

pattern histogram and a slow increase in the entropy function. Therefore, we selected the template that was determined by the MPS realization with three categories. A template with 18 conditioning points was applied to extract the spatial patterns in the downstream steps.

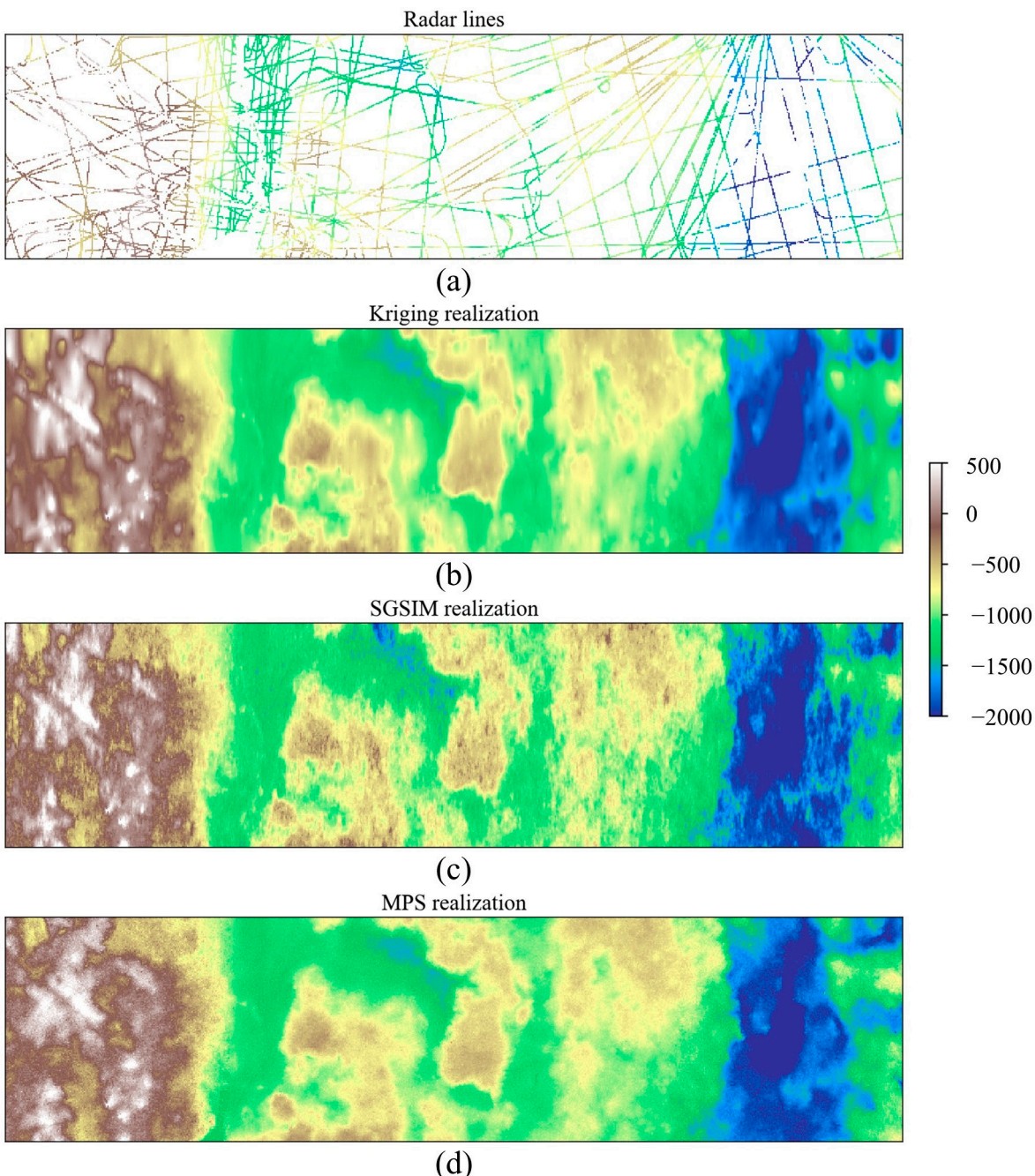

**Figure 19.** Antarctica topographic models created by the geostatistical methods. (**a**) Radar lines are the hard data in the modeling task; (**b**) Kriging realization; (**c**) the first SGSIM realization; (**d**) the first MPS realization.

Next, the stacking strategy was activated to quantify the importance of long-range and small-scale structures. With the template mentioned above, our program conducted a multi-grid analysis based on the first realization of the three geostatistical modeling methods. The grid importance is shown in Figure 21. To emphasize the key findings, we do not depict the contribution of the finest grid. It is apparent that there is a significant difference between the three realizations. On one hand, long-distance structures are not

well conserved by Kriging and SGSIM. The conditioning points collected by the large templates had a weak correlation with the template center. On the other hand, there was a strong relationship between the template center and the surrounding points in MPS realization. Compared with two-point statistics, MPS exhibits better performance in terms of long-range pattern reproduction.

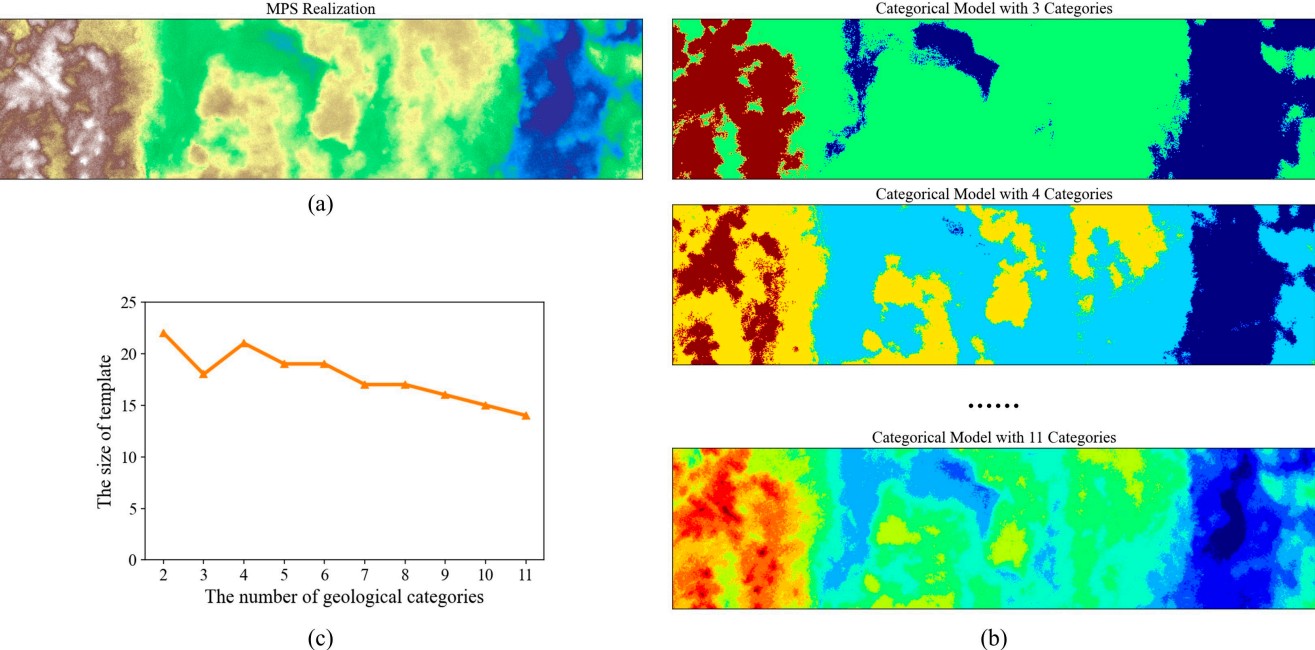

**Figure 20.** Correlation-driven template design method in the subglacial modeling case. (**a**) MPS realization; (**b**) categorical models with different numbers of geological states; (**c**) the template sizes computed by multiple categorical models.

In accordance with the adaptive template and grid importance, we performed hierarchical clustering and the decision tree classifier to characterize the subglacial models. The JS divergence and multi-dimensional scaling was carried out to quantify the morphological similarity. Figure 22a provides the evaluation results. The topographic realizations generated by MPS are highlighted in red. By contrast, blue and green are used to represent the Kriging and SGSIM models, respectively. In the feature space, two distant points indicate that there was a large mismatch between the two topographic models. Therefore, the three geostatistical methods have different behaviors in terms of their morphological characteristics. On one hand, Kriging and SGSIM are two-point-statistics modeling methods. The linear assumption is an important concept in Kriging. SGSIM applies a multi-Gaussian random function to describe spatial structures. Unknown points in SG are estimated according to a weighted combination of the surrounding points. In order to mimic the prior material, both Kriging and SGSIM utilize a variogram to allocate the weight of each conditioning point. It is challenging to reproduce geometrically complex structures in this way. On the other hand, MPS applies a template to extract spatial patterns. The relationship between the template center and the neighboring points is a core component in the simulation of geological models. Therefore, one key advantage of MPS is its ability to generate realistic realizations.

Furthermore, our PCD was used to examine the effectiveness of the DS parameter. According to the experiment conducted by Meerschman et al. [41], the DS performance largely relies on the neighboring points as well as distance tolerance. Increases in the number of neighbors have positive effects on the simulation of complicated patterns. In comparison, pattern reproduction quality can be improved by reducing the distance tolerance. Thus, we created two realization sets. First, the influence of the distance

toleration was investigated. The program fixed the neighbors $N^{DS} = 30$ and $f^{DS} = 0.1$. The tolerance $t^{DS}$ was configured as 0.025, 0.050, 0.075, and 0.100, respectively. Second, the program concentrated on the function of the neighboring points. The neighbor $N^{DS}$ varied from 10 to 40, while the $t^{DS}$ and $f^{DS}$ were specified as 0.05 and 0.1, respectively. Figure 23 displays the first realization produced by these parameter combinations.

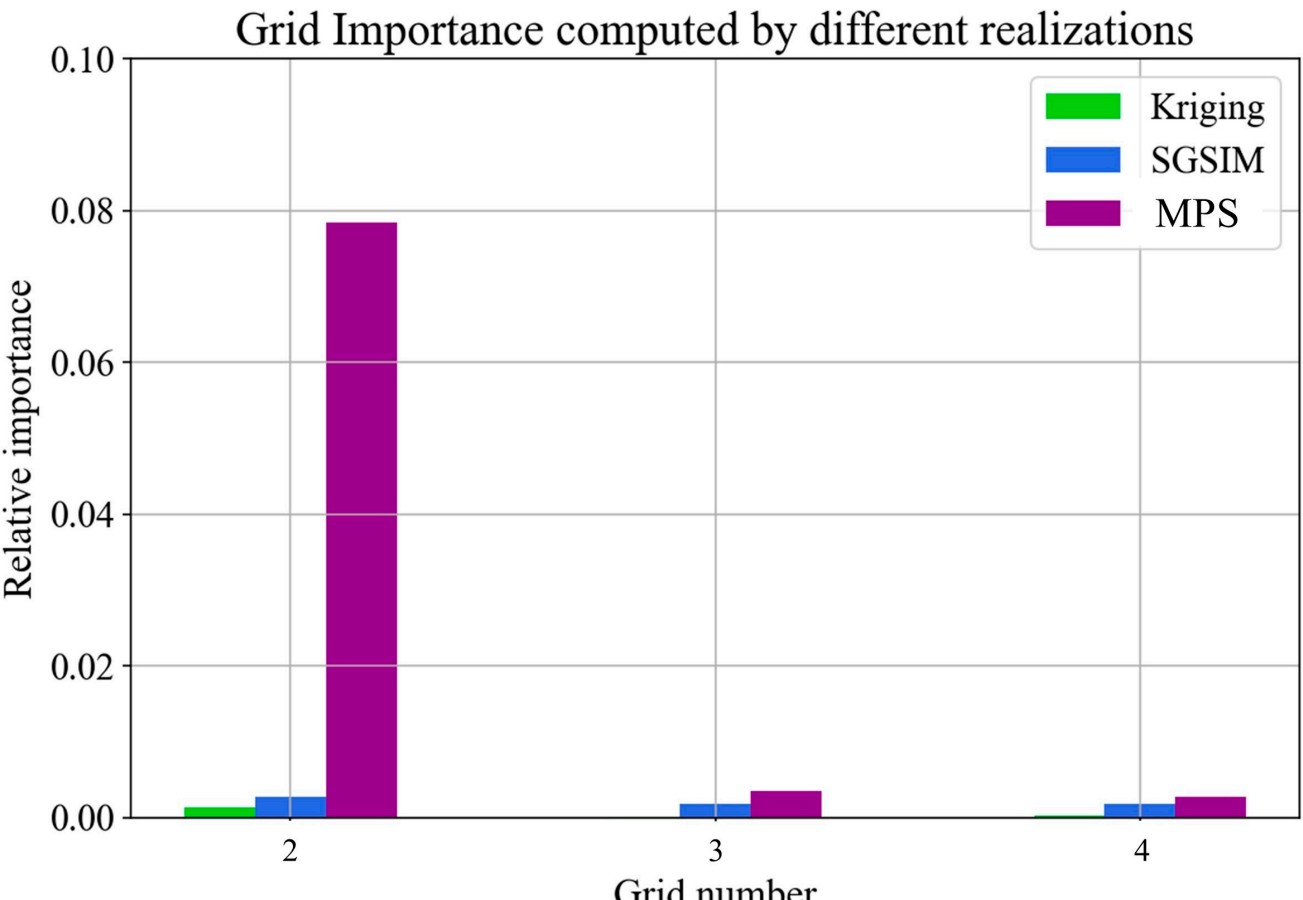

**Figure 21.** The grid importance computed by three geostatistical models.

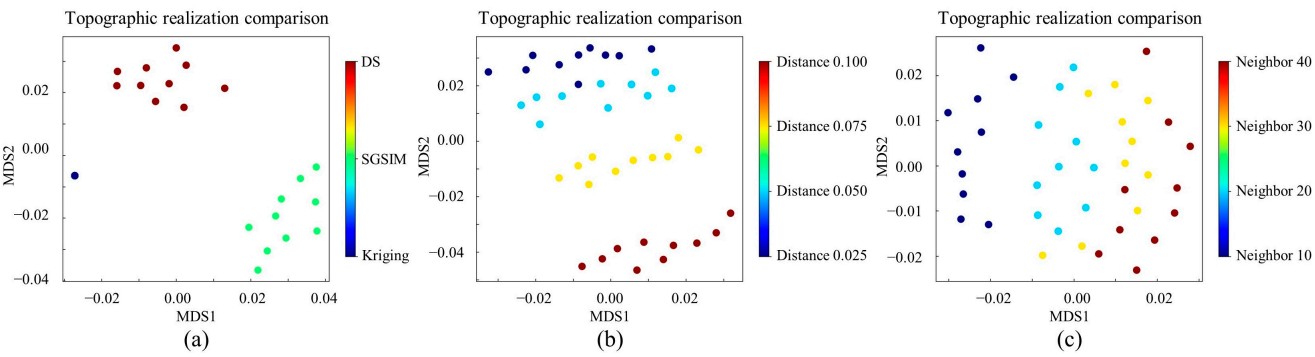

**Figure 22.** Uncertainty quantification based on 2D topographic realizations. (**a**) Model comparison between Kriging, SGSIM, and MPS; (**b**) model comparison between MPS simulations performed with different distance tolerances; (**c**) model comparison between MPS simulations performed with different neighbors.

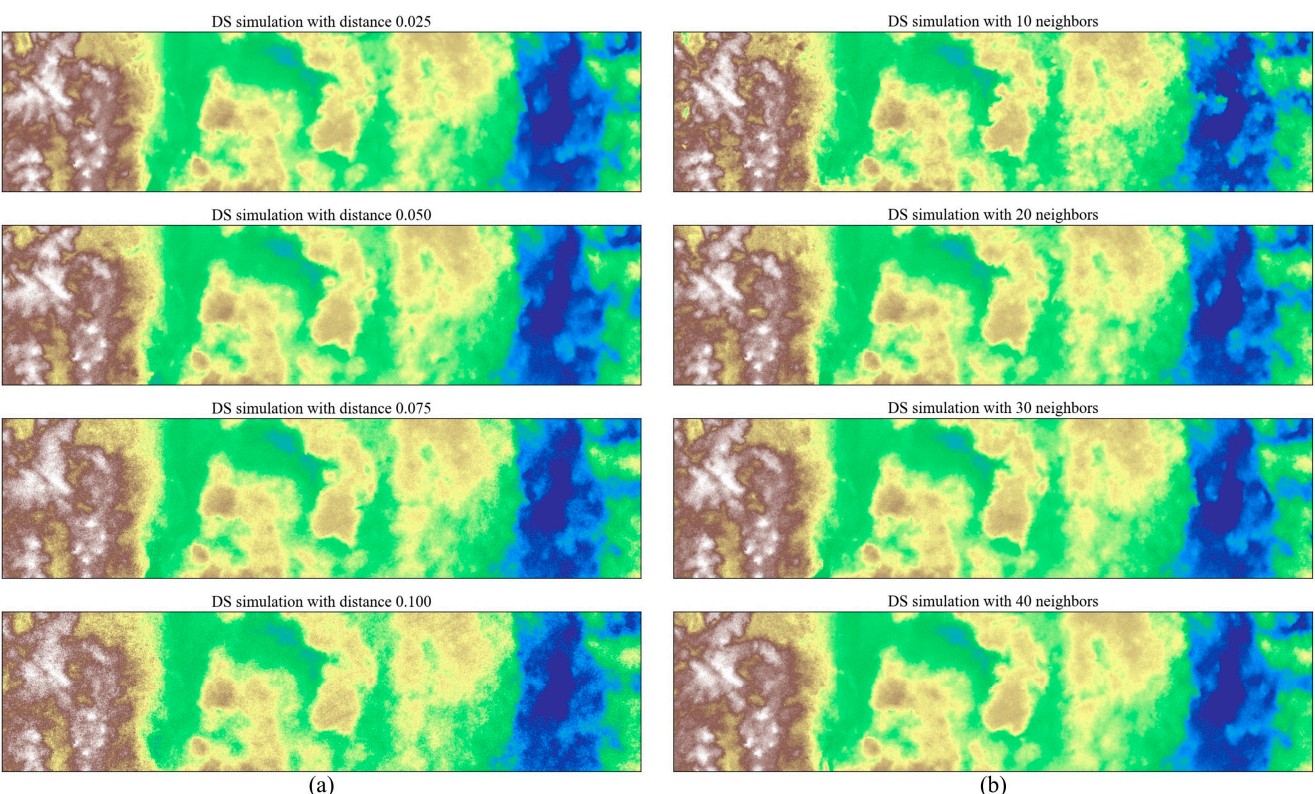

**Figure 23.** DS realizations with various parameter combinations. (**a**) Subglacial topographic models generated with different distance tolerances; (**b**) subglacial topographic models generated with different neighbors.

The proposed PCD was applied to measure the similarities between the DS models. The patterns in the first realization created by $N^{DS} = 30$, $t^{DS} = 0.05$, and $f^{DS} = 0.1$ were applied to train the decision-tree classifier. PCD calculation results are shown in Figure 22b,c. Two notable phenomena were observed. (1) The distance tolerance has a substantial effect on the modeling quality. According to Figure 22b, the red and yellow clouds had a large mismatch with other realizations. By comparison, the two blue groups were relatively close. This indicates that the topographic realizations created by $t^{DS} = 0.050$ had similar morphological characteristics to the models produced by $t^{DS} = 0.025$. However, a low value of the distance tolerance creates computational burden. Given that $t^{DS} = 0.025$, 4.07 h is necessary to create one realization. In contrast, the computer requires 0.86 h to generate a model when $t^{DS}$ is set as 0.050. Considering the time performance, 0.050 is an appropriate choice in this simulation scenario. (2) the neighboring point is a contributing factor to the simulation quality. In Figure 22c, the deep cloud is distant from the others. A small number of neighboring points in DS is not sufficient to reproduce morphologically complex structures.

### 4.4. A 3D Sandstone Model from a 2D Slice

In petroleum engineering, three-dimensional sandstone models are important materials with which to study the geometrical and physical properties of rocks [44,45]. In this section, we evaluate the PCD performance in a high-dimension sandstone system. As shown in Figure 24a, a 2D sandstone slice of $128 \times 128$ was the TI motivating a geostatistical simulation. This image was generated by computed tomography (CT) with a resolution of 10 μm. There were two geological categories. The pore and grain are expressed by the white and black areas, respectively.

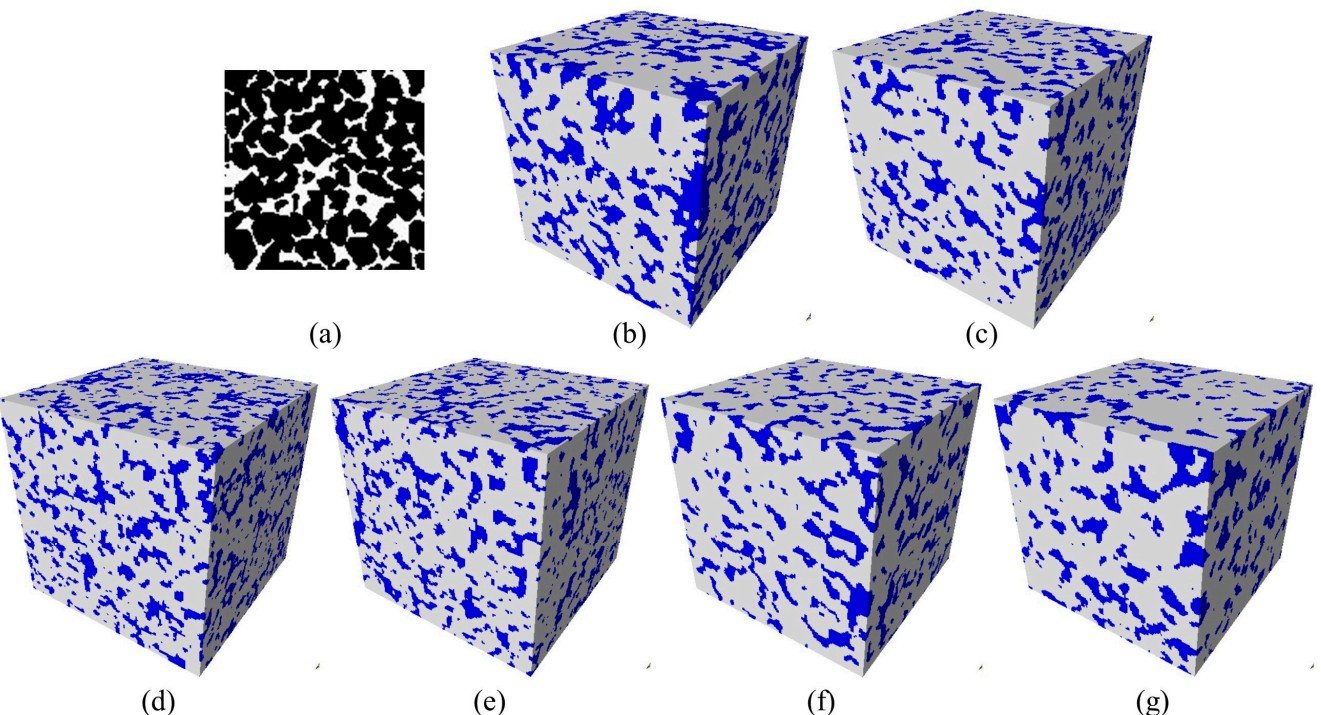

**Figure 24.** The 2D sandstone slice and 3D models. (**a**) Training image; (**b**) IMPALA realization; (**c**) NNSIM realization; (**d**) DS realization; (**e**) CDS realization; (**f**) PDFSIM realization; (**g**) SGSIM realization.

Aiming at outputting realistic models, we implemented a series of modeling programs to create 3D models on the basis of the 2D slice. First, IMPALA [38] and NNSIM [10] were activated. Because of the absence of 3D TI, these two programs were straightforwardly introduced into the probability aggregation framework. Instead of 3D patterns, the probability aggregation focuses on calculating a conditional probability based on several 2D patterns. The detailed workflow is described in [46]. We specified a template with a size of $5 \times 5$ and a multi-grid strategy with $G = 3$. Second, we implemented direct sampling (DS) [40,47] and correlation-driven direct sampling (CDS) [14]. With the intention of reproducing complex structures, CDS employed the weighted Hamming distance to define the compatible patterns. The other parameters were specified as $N^{DS} = 30$, $t^{DS} = 0.0$, and $f^{DS} = 1.0$. Third, a pattern-density-function simulation (PDFSIM) [48,49] was carried out. The core idea was to adopt the pattern density function to characterize the geological models and guide the modeling procedure. Moreover, the program designed a cascaded polymorphic method, which directly pasted a matching patch between cascaded grids within the multi-grid framework. Fourth, we performed a sequential Gaussian simulation (SGSIM) with SGeMS [37]. As a two-point statistics method, the variogram was a key tool in the description of the microstructure in the TI. Based on the parameters stated above, 10 sandstone realizations were individually created by the previous methods. The first realization is shown in Figure 24. The pore space is highlighted in blue while the grain is represented by the gray area.

The proposed PCD was launched to validate the simulation quality. To organize the spatial patterns, we specified the distance threshold as 0.1 in the hierarchical clustering. Moreover, a multi-grid strategy with $G = 4$ was adopted to capture the patterns across different scales. PCD focused on comparing IMPALA with the other modeling programs. The calculation results are shown in Figure 25.

There were three phenomena in the PCD results. (1) The CDS and PDFSIM realizations had small distances to the TI. In order to better reproduce the pore space, CDS assigned the weights of the conditioning points according to the visual features of the TI. The high correlation points had a strong influence on the MPS simulation. By comparison, PDFSIM

is an iterative program. The sandstone model is continuously upgraded until the difference from the TI is lower than the predefined threshold. (2) The high-dimension modeling strategy played an important role in the pattern reproduction as well as the spatial uncertainty. On one hand, the NNSIM and IMPALA exhibited comparable simulation qualities, as shown in Figure 25b. Although different datasets were employed, both two programs were incorporated into the probability aggregation framework. The computer employed Bordely formula to calculate the global probability according to three 2D conditional probabilities. By contrast, DS and CDS applied the majority vote to predict an unknown point in the 3D domain. There were no conditional probability computations within the DS and CDS frameworks. (3) There was a noticeable mismatch between the TI and SGSIM models. Compared with the MPS framework, the SGSIM suffers from the limited ability of the variogram. In this sandstone application, it was difficult for the two-point statistics to express the pore microstructure.

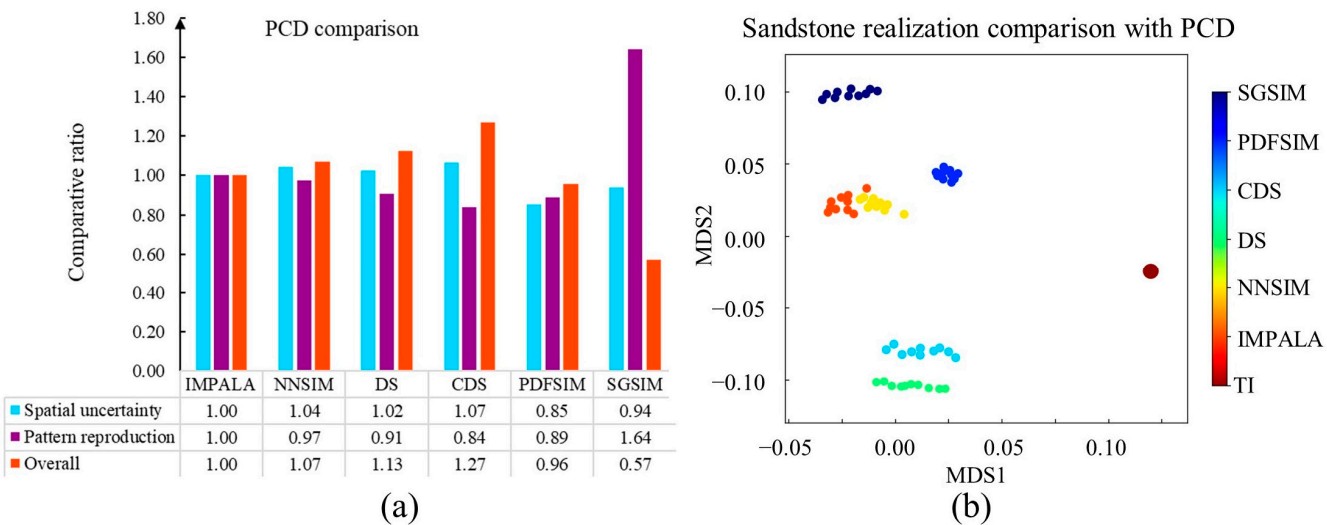

**Figure 25.** PCD evaluation of the sandstone models. (**a**) Three comparative ratios between IMPALA and other programs. (**b**) Uncertainty quantification based on sandstone models.

## 5. Discussion

As mentioned in the Introduction, pattern reproduction and spatial uncertainty are two important variabilities within geostatistical modeling. MPH focuses on recording the frequency of each pattern configuration. Pattern distributions are descriptors of geological models. By contrast, ANODI activates a clustering step to control the dimension of the pattern distribution. A multi-grid strategy is carried out to analyze spatial patterns across different scales. In this paper, we proposed a PCD method to evaluate the modeling quality and perform the uncertainty quantification. As reported in Section 4, four practical applications were conducted to examine the previous three methods. A key finding was that the performances of MPH and ANODI were heavily dependent on the parameter configuration. On one hand, the accuracy of MPH relies on the template setting and the number of grids. As shown in Figure 12b, the comparative ratio between SNESIM and DS drastically changed with the variations in the templates. With the development of the grids, the three ratios between the two realization sets approached 1.00. On the other hand, the numbers of clusters and grids were contributing factors in ANODI. According to Figures 12 and 16, the program could not output coherent results in the channel and flume simulation. The fluctuation in the comparative ratios created difficulties in the quantitative analysis of the modeling accuracy.

One key advantage of the proposed PCD is the adaptive parameter specification. In accordance with the morphological characteristics of the TI, our method automatically specifies the template configuration, the number of pattern clusters, and the importance

of each resolution. The distance threshold in the hierarchical clustering step is the only user-defined parameter. As displayed in Figure 12a, the comparison between SNESIM and DS was not significantly influenced by the value of the distance threshold. A similar phenomenon is presented in Figure 15. The comparative ratios indicated that NNSIM and TDS had comparable accuracies.

In addition, the strengths of two-point statistics and multiple-point statistics were discussed. One core concept in Kriging and SGSIM is the utilization of the variogram to explain the expected difference between two points. The simulation procedure estimates an unknown point through a weighted sum of the surrounding pixels. In contrast, MPS takes advantage of a template to explore the relationship between the template center and the neighboring points. Therefore, MPS provides a powerful way to reproduce geometrically complicated structures. As shown in Figures 19 and 21, the long-range structures were well-conserved in MPS realizations. This finding supports the investigations conducted by Yin et al. [7] and Zuo et al. [17]. According to their research, MPS has the ability to produce a group of realistic topographic models with suitable diversity. The benefits of multiple-point information are further examined in Sections 4.2 and 4.3. An extending template has a positive effect on improving the consistency between the TI and the simulated realizations. As reported in Section 4.4, we performed a 3D sandstone reconstruction. The pore microstructure was properly recreated in the MPS models. By contrast, the deep blue points corresponding to the SGSIM realizations were at large distances from the TI in Figure 25.

Next, we focused on the sensitive factors within the MPS simulation. An important finding was that the pattern similarity metric had a substantial influence on the modeling quality. In Figure 11a, the MPS programs can be divided into three groups. First, SNESIM, IMPALA, and CSSIM shared similar dispersals. The main reason is that they employ the pruning strategy to find desired pattern instances. The program removes the farthest points in the conditioning pattern if the matching pattern is not presented in the TI. Therefore, the points that are close to the template center play an important role in the pattern-searching step. Second, the Hamming distance is used by NNSIM, DS, and TDS to distinguish patterns. The template points have the same importance. The similarity between NNSIM and TDS was further validated by the flume models. Third, FILTERSIM applies six convolutional kernels to characterize 2D patterns. A compatible image patch is pasted into the simulation domain. However, one key drawback of these MPS programs is that the intrinsic characteristics of the TI are not considered in the pattern similarity metric. The contribution of each template point is fixed and constant. Accordingly, it would be interesting to assess the effect of each conditioning point during MPS simulations. Adaptive weight assignment is a promising way to further improve geostatistical modeling programs.

## 6. Conclusions

In this work, a pattern classification distribution method was proposed to assess geostatistical modeling and quantify spatial uncertainty. With the objective of improving the evaluation accuracy, a set of machine-learning techniques were employed to overcome the technical limitations in the previous multiple-point histogram and the analysis of distance methods. First, a correlation-driven template design approach was suggested to extract spatial patterns. With a region-growing program, the computer sequentially collects conditioning points according to their correlations with the template center. The number of template points is automatically determined by the elbow point of the entropy function. An irregular template of adaptive size has a positive effect on preserving the structure in the TI. Second, the proposed PCD utilizes the clustering and classification programs to characterize the geological realizations. In order to simplify the parameter setting, hierarchical clustering is launched to organize patterns in the TI. On the basis of the clustering results, a decision tree is trained to classify each pattern in geostatistical models. The program outputs a pattern distribution according to the number of member instances in each pattern category. The Jensen–Shannon divergence within the pattern

distribution becomes a measure of the similarity between two realizations. Third, a stacking framework was applied to develop the multi-grid analysis. The base classifier focuses on exploring the relationship between the template center and the neighboring points in different resolutions. By comparison, a meta-classifier was employed to evaluate the effectiveness of each base-classifier. The importance of each resolution was adaptively assigned according to the morphological characteristics of the TI.

We examined the proposed PCD by using benchmark channel models, non-stationary flume models, subglacial topographic realizations in Antarctica, and three-dimensional sandstone models. With the intention of facilitating an extensive comparison, various multiple-point statistics methods were implemented to generate geological models. The computational results indicated that our method is capable of addressing multiple geological categories, continuous variables, and high-dimensional structure. Compared with MPH and ANODI methods, the proposed PCD benefits from the automatic parameter-specification step. The underlying relationship between geostatistical realizations is efficiently recognized by our method. As the only predefined parameter, the distance threshold in the hierarchical clustering does not have a significant effect on the computational results. The findings indicate that our PCD provides a feasible way to find reliable geostatistical models and quantify spatial uncertainty.

**Author Contributions:** Conceptualization, C.Z. and Z.D.; methodology, C.Z. and Z.L.; software, C.Z. and X.W.; validation, X.W. and Y.W.; writing—original draft preparation, C.Z. and Z.L.; writing—review and editing, Z.D., X.W. and Y.W. All authors have read and agreed to the published version of the manuscript.

**Funding:** This research was funded by Natural Science Foundation of Shaanxi Province, grant number 2022JQ-227; the Postdoctoral Science Foundation of China, grant number 2022M710482; and the Department of Transportation Science and Technology Project of Zhejiang Province, grant number 2023016.

**Data Availability Statement:** All data generated or analyzed during this study are included in this published article.

**Conflicts of Interest:** The authors declare no conflict of interest.

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
