# Peer review of "A Pattern Classification Distribution Method for Geostatistical Modeling Evaluation and Uncertainty Quantification"

_remotesensing, doi:10.3390/rs15112708_

Round 1

Reviewer 1 Report

This paper proposes a pattern classification distribution to compare geostatistical realizations, clustering and classification methods to characterize the geological models, and finally, a framework is in charge of multi-grid analysis. The paper addresses challenges in the context of geostatistical modeling, is easy to read, and is well-referenced. I only have some minors comments that could improve the quality of the paper:

- The structure of the paper is missing at the end of the introduction.

- The quality of all figures should be definitely improved.

- Change the title of section 2 to Background.. Actually, the related work is distributed throughout the whole article instead of being in the related work section like most articles. I liked the way you provided in this paper.

Author Response

Responses to the Comments from the Reviewer 1

Comment 1: This paper proposes a pattern classification distribution to compare geostatistical realizations, clustering and classification methods to characterize the geological models, and finally, a framework is in charge of multi-grid analysis. The paper addresses challenges in the context of geostatistical modeling, is easy to read, and is well-referenced. I only have some minors comments that could improve the quality of the paper:

Response 1:

First of all, we are grateful to the referee for the review. Each comment is insightful and has a positive effect on improving our manuscript.

Comment 2: - The structure of the paper is missing at the end of the introduction.

Response 2:

Based on this valuable comment, we add a new paragraph at the end of the Introduction. The related content is shown below.

“The rest of this paper is organized as follows. Section 2 establishes the context of the geostatistical evaluation methods and provides detailed procedures within MPH and ANODI. Our proposed PCD is explained in Section 3. Section 4 presents four real-world applications. The experiment results and findings are discussed in Section 5. Finally, conclusions are drawn in Section 6.”

Comment 3: The quality of all figures should be definitely improved.

Response 3:

We appreciate the reviewer for this suggestion. The quality of figures is an important factor to ensure the quality of our manuscript. In the last five days, we recreated all figures. To provide a clear print quality, the dots per inch (DPI) increases from 150 to 300. New figures are inserted in the manuscript and uploaded to the submission system.

Moreover, we modify Figure 11 with the objective to distinguish geostatistical programs. Three multiple-point statistics (MPS) realization groups are independently displayed in the right column of each subfigure. The new description in Section 4.1 is shown in the following.

“Furthermore, we activate multi-dimensional scaling (MDS) to visualize the calculation results. In the feature space, each node represents a geological model. Two close points imply that there is an intensive compatibility between two MPS realizations. Figure 11(a) displays the MDS visualization result. To avoid visual confusion, we partition the point cloud into three parts. First, SNESIM, IMPALA and CSSIM realizations are emphasized. Second, the blue points display the dispersals of NNSIM, DS and TDS models. Third, FILTERSIM realizations are presented in yellow.”

Comment 4: Change the title of section 2 to Background.. Actually, the related work is distributed throughout the whole article instead of being in the related work section like most articles. I liked the way you provided in this paper.

Response 4:

Thanks for the attentive reading. We modify the title of Section 2 on the basis of this excellent advice. The new content is exhibited as follows:

“2. Background of the Geostatistical Evaluation Methods”

Please see the attachment for the revision summary and updated figures.

Reviewer 2 Report

Accurate geostatistical modelling is of great significance for development of subsurface resources and prediction of surficial geomorphology. Pattern recognition and classification is always a central part for geostatistical modelling. This manuscript propose a new pattern classification distribution method to assess the geomodelling process and results. From the several application cases, I can see the proposed method truly outperforms other traditional ones, even in the 3D digital rock reconstruction cases.
Generally, the research is pretty solid, the logic is well presented, the English language is good, and figures are also presenting important contents of the manuscript.

Here, I only have several minor questions / comments:

1. In the paragraph of line 63 – 73, except honoring expected geological patterns and producing multiple realizations to represent implicit uncertainty of geomodels, another very important aspect of geomodelling is conditioning, i.e., to let the produced models condition to given sparse borehole interpretation, geophysical data, and possibly well production data. Thus, such a point may need to be supplemented into this paragraph.

2. In Introduction part, most geological pattern-related contents in geostatistical discipline are reviewed and well presented. Only one point may need to be further discussed:

The essence of GANs-based geomodelling is also to capture complex spatial geological patterns using neural networks and then to apply the captured patterns to produce realistic geomodels. Therefore, in the context of this manuscript, when introducing geological patterns, GANs-based geomodelling methods and pattern capturing manner are also suggested to be supplemented.

Laloy, E., Hérault, R., Jacques, D., & Linde, N. (2018). Training-image based geostatistical inversion using a spatial Generative Adversarial Neural Network. Water Resources Research, 54(1), 381–406. https://doi.org/10.1002/2017WR022148

Song, S., Mukerji, T., & Hou, J. (2021a). Geological facies modeling based on progressive growing of Generative Adversarial Networks (GANs). Computational Geosciences. https://doi.org/https://doi.org/10.1007/s10596-021-10059-w

Zhang, T. F., Tilke, P., Dupont, E., Zhu, L. C., Liang, L., & Bailey, W. (2019). Generating geologically realistic 3D reservoir facies models using deep learning of sedimentary architecture with generative adversarial networks. Petroleum Science, 16(3), 541–549. https://doi.org/10.1007/s12182-019-0328-4

3. When describing the downsampling of grids (Fig. 3), it is important to briefly introduce the downsampling approaches, e.g., facies frequency based.

4. In Fig 11, there are too many categories in each sub-map. Is it possible to only present up to 4 categories on each map? Of course, more maps can be presented. E.g., map 1 shows results of method A are within the distribution of results of method B, and map 2 shows results of method B are almost overlap with the distribution of results of method C. 

Author Response

Responses to the Comments from the Reviewer 2

Comment 1: Accurate geostatistical modelling is of great significance for development of subsurface resources and prediction of surficial geomorphology. Pattern recognition and classification is always a central part for geostatistical modelling. This manuscript propose a new pattern classification distribution method to assess the geomodelling process and results. From the several application cases, I can see the proposed method truly outperforms other traditional ones, even in the 3D digital rock reconstruction cases.
Generally, the research is pretty solid, the logic is well presented, the English language is good, and figures are also presenting important contents of the manuscript.

Response 1:

Prior to the point-by-point response, we would like to show our gratitude to the reviewer for his/her careful reading and valuable suggestions. Each comment plays an essential role in the manuscript revision.

Comment 2: Here, I only have several minor questions / comments:

  1. In the paragraph of line 63 – 73, except honoring expected geological patterns and producing multiple realizations to represent implicit uncertainty of geomodels, another very important aspect of geomodelling is conditioning, i.e., to let the produced models condition to given sparse borehole interpretation, geophysical data, and possibly well production data. Thus, such a point may need to be supplemented into this paragraph.

Response 2:

The reviewer is very insightful about the geological modeling program. Besides pattern reproduction and spatial uncertainty, observation conditioning is a key aspect of the modeling quality. Since it is directly sampled from the investigated area, borehole data becomes a hard constraint to the geostatistical simulation program. By comparison, the geophysical map produced by ground penetrating radar focuses on explaining the trend of the geological structure. Therefore, the multiple-point statistics (MPS) program regards the geophysics data as the soft data during the simulation.

Based on this valuable comment, we add several new sentences in the Introduction section. The related content is shown in the following.

“Furthermore, the observation variable is important prior knowledge in the conditional simulation scenario. For example, the borehole interpretation is directly sampled from the subsurface system [17]. Produced by ground penetrating radar, the geophysical data describes the trend of the geological structure under investigation [30]. It is necessary to respect the conditioning data during geological modeling. These conflicting objectives bring a challenge to the simulation program.”

Comment 3: In Introduction part, most geological pattern-related contents in geostatistical discipline are reviewed and well presented. Only one point may need to be further discussed:

The essence of GANs-based geomodelling is also to capture complex spatial geological patterns using neural networks and then to apply the captured patterns to produce realistic geomodels. Therefore, in the context of this manuscript, when introducing geological patterns, GANs-based geomodelling methods and pattern capturing manner are also suggested to be supplemented.

Laloy, E., Hérault, R., Jacques, D., & Linde, N. (2018). Training-image based geostatistical inversion using a spatial Generative Adversarial Neural Network. Water Resources Research, 54(1), 381–406. https://doi.org/10.1002/2017WR022148

Song, S., Mukerji, T., & Hou, J. (2021a). Geological facies modeling based on progressive growing of Generative Adversarial Networks (GANs). Computational Geosciences. https://doi.org/https://doi.org/10.1007/s10596-021-10059-w

Zhang, T. F., Tilke, P., Dupont, E., Zhu, L. C., Liang, L., & Bailey, W. (2019). Generating geologically realistic 3D reservoir facies models using deep learning of sedimentary architecture with generative adversarial networks. Petroleum Science, 16(3), 541–549. https://doi.org/10.1007/s12182-019-0328-4

Response 3:

We sincerely appreciate the reviewer for his/her recommendation. In recent years, the development of deep neural network has gained considerable attention. Aiming at establishing an extensive background in the Introduction, we expand the description of geological modeling programs. The new sentences are displayed as follows.

“In addition, the development of generative adversarial network (GAN) technique has accepted considerable attention in the geostatistics community [20, 21]. Based on a large amount of TIs, two neural networks are simultaneously trained through an adversarial competition. A generator network attempts to produce an image associated with similar characteristics to TIs. By contrast, the discriminator is responsible for distinguishing real and simulated models. The expanding applications of GAN include geological facies [22, 23], probability inversion [24] and porous media [25].”

Accordingly, the relevant articles are supplemented in the Reference section.

“20. Song, S.; Mukerji, T.; Hou, J. Bridging the gap between geophysics and geology with Generative Adversarial Networks (GANs). IEEE Trans. Geosci. Remote. Sens. 2022, 60, 1-11.

21. Li, X.; Li, B.; Liu, F.; Li, T.; Nie, X. Advances in the application of deep learning methods to digital rock technology. Adv. Geo-Energy Res. 2022, 8(1): 5-18.

22. Song, S.; Mukerji, T.; Hou, J. GANSim: Conditional facies simulation using an improved progressive growing of Generative Adversarial Networks (GANs). Math. Geosci. 2021, 53, 1413–1444.

23. Zhang, T. F.; Tilke, P.; Dupont, E.; Zhu, L. C.; Liang, L.; Bailey, W. Generating geologically realistic 3D reservoir facies models using deep learning of sedimentary architecture with generative adversarial networks. Pet. Sci., 2019, 16, 541–549.

24. Laloy, E.; Hérault, R.; Jacques, D.; Linde, N. Training-image based geostatistical inversion using a spatial Generative Adversarial Neural Network. Water Resour. Res. 2018, 54(1), 381–406.

25. Chen, Q.; Cui, Z.; Liu, G.; Yang, Z.; Ma, X. Deep convolutional generative adversarial networks for modeling complex hydrological structures in Monte-Carlo simulation, J. Hydrol., 2022, 610, 127970. ”

Comment 4: When describing the downsampling of grids (Fig. 3), it is important to briefly introduce the downsampling approaches, e.g., facies frequency based.

Response 4:

We are grateful for this suggestion. In the multi-grid strategy, downsampling is a key step to create the pyramid of multi-resolution views. With the aim of avoiding confusion, we supplement a brief explanation in Section 2.2. The new sentences are shown below.

“As Figure 3 shows, the program creates a pyramid of multi-resolution views. Inspired by MPS, the coarse grid is recursively generated by subsampling the fine grid. In this conceptual case, we implement a down-sampling procedure of stride 2. Starting from the bottom-left corner, the pixels in the fine grid are sequentially checked. The computer removes every even-numbered row and column to produce a small grid. For the complex scenario, the Gaussian pyramid and facies frequency-based method are favorable to preserve important geological structures.”

Comment 5: In Fig 11, there are too many categories in each sub-map. Is it possible to only present up to 4 categories on each map? Of course, more maps can be presented. E.g., map 1 shows results of method A are within the distribution of results of method B, and map 2 shows results of method B are almost overlap with the distribution of results of method C.

Response 5:

We appreciate the reviewer for providing this suggestion. Figure 11 is regenerated to present the visualization result calculated by multi-dimensional scaling. Three multiple-point statistics (MPS) groups are individually expressed. The maximum number of colors in each subfigure is limited to four. Moreover, we provide sentences to explain the updated figure in Section 4.1. The related content is listed as follows.

“Furthermore, we activate multi-dimensional scaling (MDS) to visualize the calculation results. In the feature space, each node represents a geological model. Two close points imply that there is an intensive compatibility between two MPS realizations. Figure 11(a) displays the MDS visualization result. To avoid visual confusion, we partition the point cloud into three parts. First, SNESIM, IMPALA and CSSIM realizations are emphasized. Second, the blue points display the dispersals of NNSIM, DS and TDS models. Third, FILTERSIM realizations are presented in yellow.”

Please see the attachment for the revision summary and updated figures.
